# Membrane depolarization kills dormant *Bacillus subtilis* cells by generating a lethal dose of ROS

Declan A. Gray[1,2,7,9], Biwen Wang [3,9], Margareth Sidarta [2,4], Fabián A. Cornejo [5], Jurian Wijnheijmer [3], Rupa Rani[2,4], Pamela Gamba[1,8], Kürşad Turgay [5,6], Michaela Wenzel [2,4], Henrik Strahl [1] & Leendert W. Hamoen [1,3] ✉

The bactericidal activity of several antibiotics partially relies on the production of reactive oxygen species (ROS), which is generally linked to enhanced respiration and requires the Fenton reaction. Bacterial persister cells, an important cause of recurring infections, are tolerant to these antibiotics because they are in a dormant state. Here, we use *Bacillus subtilis* cells in stationary phase, as a model system of dormant cells, to show that pharmacological induction of membrane depolarization enhances the antibiotics' bactericidal activity and also leads to ROS production. However, in contrast to previous studies, this results primarily in production of superoxide radicals and does not require the Fenton reaction. Genetic analyzes indicate that Rieske factor QcrA, the iron-sulfur subunit of respiratory complex III, seems to be a primary source of superoxide radicals. Interestingly, the membrane distribution of QcrA changes upon membrane depolarization, suggesting a dissociation of complex III. Thus, our data reveal an alternative mechanism by which antibiotics can cause lethal ROS levels, and may partially explain why membrane-targeting antibiotics are effective in eliminating persisters.

Many commonly used antibiotics that interfere with DNA synthesis, such as ciprofloxacin, or protein synthesis, such as kanamycin and gentamicin, or cell wall synthesis, such as vancomycin and ampicillin, trigger the production of lethal levels of reactive oxygen species (ROS), which contributes to the bactericidal activity of these antibiotics[1–4]. The produced ROS consists primarily of hydroxyl radicals generated by the Fenton reaction. The free ferrous iron required for this reaction is assumed to originate from iron-sulfur proteins by a yet not fully understood mechanism[1–3]. Despite this double mode of action, bacterial cells can tolerate these antibiotics by shutting down biosynthetic processes and entering a physiologically dormant state[5,6]. This so-called 'persister' state can be caused by activation of toxin-antitoxin systems[6,7], the stringent response[8], SOS response[9,10] or simple nutrient starvation[11]. Persister cells can resume growth when the antibiotic treatment ceases[6], and are therefore believed to be an important source of recurrent infections[12,13]. Therapeutic strategies for eradicating persister cells are in dire need.

[1]Centre for Bacterial Cell Biology, Biosciences Institute, Faculty of Medical Sciences, Newcastle University, Baddiley-Clark Building, Newcastle upon Tyne NE2 4AX, UK. [2]Centre for Antibiotic Resistance Research in Gothenburg (CARe), Gothenburg, Sweden. [3]Swammerdam Institute for Life Sciences, University of Amsterdam, Science Park 904, C3.108, 1098 XH Amsterdam, The Netherlands. [4]Division of Chemical Biology, Department of Life Sciences, Chalmers University of Technology, Kemigården 4, 412 96 Gothenburg, Sweden. [5]Max Planck Unit for the Science of Pathogens, Charitéplatz 1, 10117 Berlin, Germany. [6]Leibniz Universität Hannover, Institut für Mikrobiologie, Herrenhäuser Str. 2, 30419 Hannover, Germany. [7]Present address: Department of Infectious Diseases, Institute of Biomedicine, The Sahlgrenska Academy at the University of Gothenburg, Gothenburg, Sweden. [8]Present address: Charles River Laboratories, Keele Science Park, Keele ST5 5SP, UK. [9]These authors contributed equally: Declan A. Gray, Biwen Wang. ✉e-mail: l.w.hamoen@uva.nl

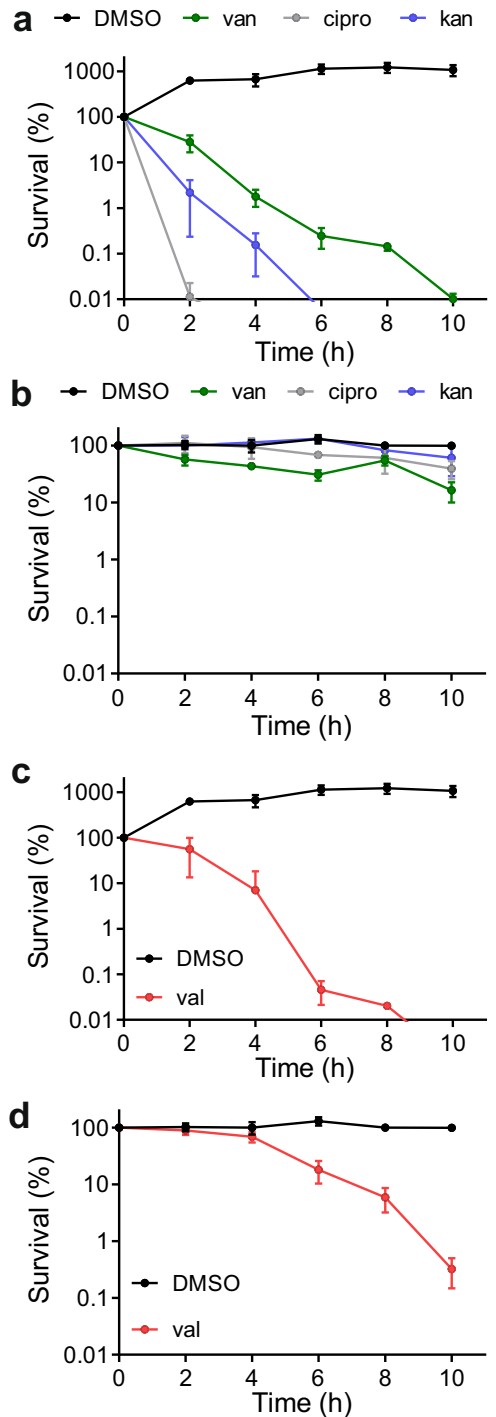

**Fig. 1 | Antibiotic killing of non-growing *B. subtilis* cells.** Survival curves of exponentially growing (**a** and **c**), and stationary (dormant) phase (**b** and **d**) *B. subtilis* cells incubated with different antibiotics. Cells were treated with either 20 μg/ml vancomycin (van), 2 μg/ml ciprofloxacin (cipro), 10 μg/ml kanamycin (kan) or 100 μM valinomycin (val). 1 % DMSO (w/w) was used as control. *B. subtilis* strains contained a deletion in *spoIIE* and were unable to sporulate. Percent survival was determined through plating serial dilutions and viable count measurements every 2 h. Data shown reflect mean ± SD (standard deviation) of three biological replicates.

We previously showed that the membrane potential is essential for the cellular localization of different bacterial peripheral membrane proteins, such as the conserved cell division proteins FtsA and MinD[14]. Persister cells do not grow, but it is likely that they maintain a membrane potential. Presumably, dissipation of the membrane potential will also result in the delocalization of membrane-associated proteins in persister cells, which could affect their viability. We have investigated this possibility by using dormant, antibiotic-tolerant, *Bacillus subtilis* cells as a simple model system for persisters. Surprisingly, we found that dissipation of the membrane potential generates lethal levels of reactive oxygen species (ROS). This finding was counterintuitive since ROS production is generally linked to enhanced respiration[15], and in fact membrane depolarization has been shown to reduce electron transfer in the respiratory chain of *B. subtilis*[16]. Moreover, we found that the Fenton reaction was not involved and that especially superoxide radicals were formed. By means of genetic analysis, we could pinpoint one of the sources of ROS to the conserved iron-sulfur cluster protein QcrA of respiratory complex III, also known as Rieske factor[17]. Interestingly, microscopic analysis showed that QcrA delocalizes after membrane depolarization, possibly indicating that detachment of QcrA from complex III leads to superoxide radical formation. This killing mechanism may explain why membrane-targeting compounds are successful in eradicating antibiotic-tolerant persister cells[18–22]

## Results

### Antibiotic-tolerant *B. subtilis* cells

The antibiotic tolerance of persisters ensues from the fact that these cells are in a metabolically dormant state[6,7]. It has been shown that *Escherichia coli* cells in the stationary growth phase display an antibiotic tolerance that is reminiscent of persister cells[23]. To examine whether the same is true for *B. subtilis*, we used a sporulation-deficient mutant Δ*spoIIE*, since *B. subtilis* spores are insensitive to antibiotics and, thus, would confound the measurements. The Δ*spoIIE* deletion mutant was grown overnight (18 h) in Lysogeny Broth (LB) medium to stationary phase, and subsequently treated with 10 x minimal inhibitory concentrations of either vancomycin, kanamycin, or ciprofloxacin for 10 h (see Table S4 for MICs). Samples for colony forming unit (CFU) measurements were taken every 2 h, and an exponentially growing culture was used as non-dormant comparison. As shown in Fig. 1, the latter culture was sensitive for all three antibiotics whereas the overnight culture was not. Of note, when exponentially growing cells were washed and incubated for 2 h in Phosphate Buffered Saline (PBS) buffer to stop growth, they did not become resistant to these antibiotics and remained sensitive. In summary, non-sporulating stationary phase *B. subtilis* cells can be used as a simple model system for antibiotic tolerant persister cells.

These stationary phase cells maintained membrane potential levels comparable to actively growing cells (Fig. S1a). Many peripheral membrane proteins use an amphipathic alpha helix domain as a reversible membrane anchor and we have shown that the binding of such an anchor domain is strongly stimulated by the membrane potential[14]. Depolarization of the membrane potential might therefore disturb the normal localization of different peripheral membrane proteins, which in turn, could affect the viability of persister cells. To test whether dormant *B. subtilis* cells are sensitive for membrane depolarization, we exposed them to the potassium ionophore valinomycin, which specifically dissipates the transmembrane potential[24,25] (Fig. S1). To assure full dissipation over a 10 h incubation period, we used 100 μM valinomycin, which is 10x the MIC for exponentially growing cells (Table S4). Membrane depolarization by valinomycin requires the presence of sufficient potassium ions in the medium[25]. Therefore, 300 mM KCl was included in the medium, as well as 50 mM HEPES. This buffer was added to ensure a stable pH throughout the cultivation, which is important since the activity of valinomycin decreases upon acidification of the culture[26]. For the wild type and several mutants used in this study, we checked whether the pH remained constant during the 10 h incubation with valinomycin and this was indeed the case (Fig. S2). The addition of valinomycin reduced

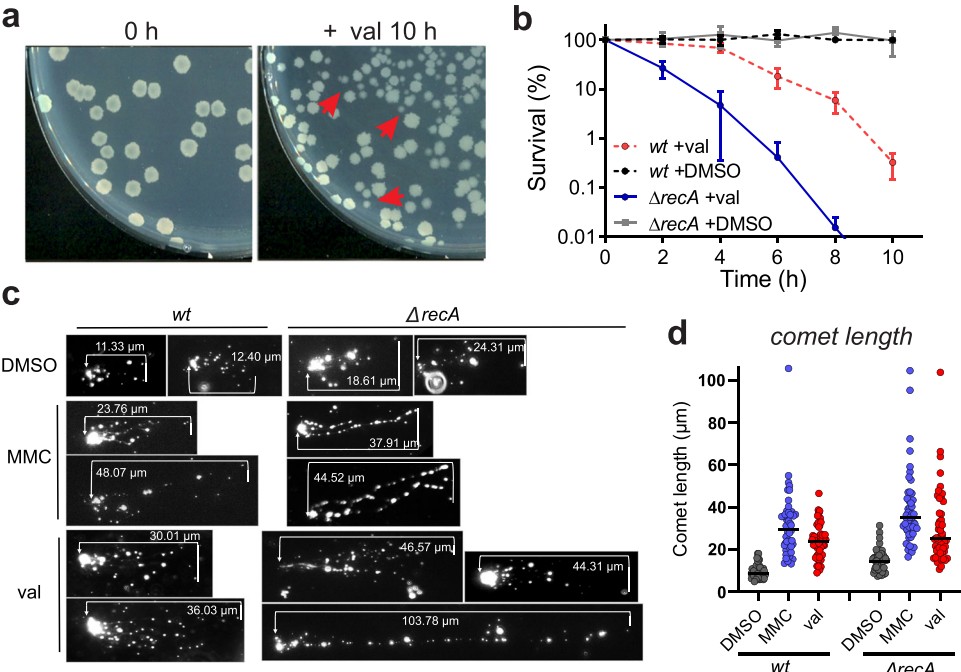

**Fig. 2 | Possible cause of killing upon membrane potential dissipation.**
**a** Colonies of stationary phase cells treated with valinomycin for 0 h and 10 h. Some small colonies are indicated by red arrows. **b** Survival curves of a *recA* deletion strain (Δ*recA*). 100 μM valinomycin was used and 1 % DMSO as negative control. *wt* indicates wild type cells. The data shown reflect the mean ± SD of three biological replicates. **c** DNA damage assessment using a comet assay. Wild type (*wt*) and Δ*rec*A cells were treated 2 h with either 100 μM valinomycin (val) or 50 ng/ml of the DNA damaging agent mitomycin C (MMC). Control cells were treated with 1 % DMSO for 2 h. Cells were lysed and electrophoresed on microscopy slides, and nucleoids were visualized using fluorescence light microscopy. Increased DNA breakage results in more stretched DNA. **d** Quantification of the DNA comet lengths of 50 cells.

the viable count of exponentially growing cells by 90 % after 2 h incubation (Fig. 1c). The dormant culture showed some degree of resilience compared to its actively growing counterparts, but after some time the viability decreased and after 6 h approximately 90 % of the cells were killed (Fig. 1d), suggesting that membrane-depolarizing compounds might indeed be useful to combat persisters. The efficacy of antibiotics can be influenced by the concentration of cells[27]. To test whether this is also the case for valinomycin, we followed the killing of 10x concentrated exponentially growing cultures and 10x diluted stationary phase cultures, and found that higher concentrations of cells moderately reduce the activity of valinomycin (compare Fig. 1c, d and Fig. S3), which might be the reason for the higher sensitivity of exponentially growing cells compared to stationary phase cells. To prevent cell concentration effects, the cell (optical) density was kept the same in the experiments.

## Cause of killing
The killing kinetic of valinomycin is quite different between exponentially growing and dormant cells, and in the latter case showed a clear acceleration over time (Fig. 1c, d). This made us wonder why dormant cells eventually succumb when their membrane is depolarized. It has been shown that depolarization of the *B. subtilis* cell membrane results in an uncontrolled autolysin activity and cell lysis[28]. To determine whether this could be responsible for the killing of dormant cells, we exposed a mutant lacking all major autolysins (Δ*lytA*, Δ*lytB*, Δ*lytC*, Δ*lytD*, Δ*lytE* and Δ*lytF*) to valinomycin. However, this deletion mutant did not show an increase in viability (Fig. S4a). *E. coli* persisters can be killed by the induction of cryptic prophages[29]. However, a *B. subtilis* strain cured of all prophages exhibited a similar sensitivity for valinomycin as the wild type strain (Fig. S4b), indicating that the reduction in viable count is also not related to prophage activation.

During the viable count measurements, we noticed that after prolonged valinomycin treatment the colony size started to vary and smaller colonies emerged (Fig. 2a). Such variation in colony size is also observed when bacterial cells are treated with DNA-damaging agents[30], suggesting that membrane depolarization might cause DNA damage. If this is true, then a DNA repair mutant, such as a Δ*recA* mutant, should be more sensitive to valinomycin, and this was indeed the case (Fig. 2b).

The Δ*recA* mutant grows slightly slower compared to the wild-type strain, but reaches comparable cell density in the stationary phase and does not display increased cell lysis (Fig. S5), suggesting that the sensitivity of this mutant for valinomycin is not caused by a reduced stationary phase carrying capacity. To confirm the DNA damage directly, we explored the use of the comet assay, a technique commonly used to assess DNA breakage in eukaryotic cells[31]. In this assay, cells are mixed with agarose on a microscope slide, and after lysis exposed to an electrophoretic field, stretching the chromosomal DNA into a comet-like structure. The resulting tail will stretch further when more DNA breaks are present. As shown in Fig. 2c, d, incubation with valinomycin increased the comet tail, almost as much as the DNA-damaging agent mitomycin C. When RecA was absent, the comet tails became longer (Fig. 2c, d). These data strongly suggest that valinomycin affects viability by introducing DNA damage.

## Generating ROS
A common cause of DNA damage is the accumulation of reactive oxygen species (ROS) in the cell. However, it seems unlikely that dormant *B. subtilis* cells would produce ROS upon membrane depolarization, since the accumulation of ROS is normally associated with hyper respiration, which can be prevented by membrane potential dissipating agents[16,32,33]. Moreover, antibiotics that have been shown to generate ROS, including norfloxacin, vancomycin and kanamycin[1–3], are not active in dormant cells. ROS is primarily a by-product of aerobic respiration and under anaerobic conditions ROS levels are generally much lower[34]. When dormant *B. subtilis* cells were placed into an anaerobic chamber for 10 h only a fraction of cells survived (Fig. 3a).

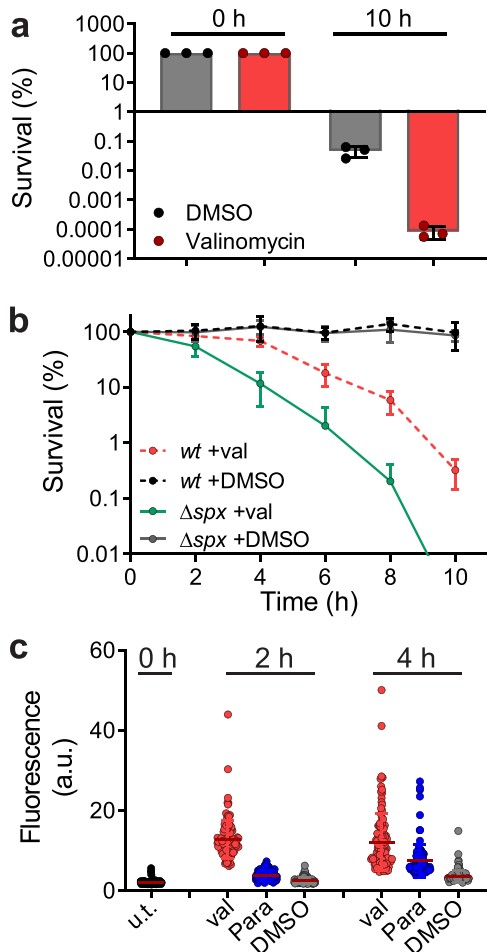

**Fig. 3 | Generation of ROS. a** Survival of stationary phase cells under anaerobic conditions in the absence or presence of 100 µM valinomycin. The addition of 1 % DMSO was used as control. Viable counts were determined after 0 and 10 h incubation. Data reflect mean ± SD of three biological replicates. **b** Survival curves of stationary phase wild type *(wt)*, Δ*recA* and Δ*spx* cells in the presence of 100 µM valinomycin (val) or 1 % DMSO. Data shown reflect mean ± SD of three biological replicates. The viable counts of Δ*recA* and Δ*spx* cells in the presence of 1 % DMSO were similar to that of wild-type cells and are not indicated here. **c** ROS production in stationary phase cells measured by the fluorescent ROS probe H2DCFDA. Cells were treated with either 100 µM valinomycin, 1 mM paraquat, or 1 % DMSO for 2 and 4 h. The fluorescence intensities of 120 cells were measured microscopically and plotted. Three biological replicates were performed to ensure repeatability.

Although *B. subtilis* can grow anaerobically, it requires certain medium conditions and sufficient time to adapt[35], which likely explains this reduction in viability. Nevertheless, the fraction of surviving cells was still smaller when valinomycin was added (Fig. 3a).

Since the viability of *B. subtilis* is greatly affected by anaerobic conditions, even in the absence of valinomycin, this experiment did not reveal whether the DNA damage caused by valinomycin treatment was due to the accumulation of ROS. Therefore, we took another approach and tested the sensitivity of a *spx* deletion mutant. Spx is the key regulator of the oxidative stress response in *B. subtilis* and is required for survival in the presence of strong ROS-inducing compounds such as paraquat[36]. Indeed, a Δ*spx* deletion mutant was considerably more sensitive to valinomycin (Fig. 3b), supporting the idea that ROS is a key contributor to the killing of depolarized cells. Finally, to directly show that ROS was generated, we used the fluorescent ROS probe 2',7'-dichlorodihydrofluorescein diacetate (H2DCFDA)[3]. Dormant *B. subtilis* cells were incubated with H2DCFDA and exposed to either valinomycin or the well-known ROS inducer paraquat for 2 and 4 h[37]. The increase in

fluorescence in cells was measured using fluorescence light microscopy. As shown in Fig. 3c, incubation with valinomycin for 2 h resulted in a strong increase in ROS that remained high over a 4 h period and even exceeded the effect of 1 mM paraquat. To confirm these results, we tested another ROS probe, Oxyburst green, which gave comparable results (Fig. S6). Thus, membrane depolarization of dormant *B. subtilis* cells does lead to the accumulation of ROS.

## Determining ROS type

Hydroxyl radicals ($^\bullet$OH) and superoxide radicals ($O_2^{\bullet-}$) are the main reactive oxygen species formed during aerobic growth[38]. Incubation with antibiotics like norfloxacin, ampicillin, and kanamycin results primarily in the formation of hydroxyl radicals made from hydrogen peroxide by the Fenton reaction[1–3]. This conclusion was based on the fact that $Fe^{2+}$-specific chelators like ferrozine or bipyridyl and the hydroxyl radical scavenger thiourea reduced the formation of radicals and diminished the killing efficiency of these antibiotics[1–3]. Moreover, increased levels of catalase, which removes hydrogen peroxide[39], also suppressed the killing by either ampicillin, gentamicin, or norfloxacin[3]. Interestingly, the addition of 0.5 mM ferrozine had no effect on the viability of dormant cells when incubated with valinomycin, and 0.5 mM bipyridyl even reduced the viability (Fig. 4a). Moreover, the presence of 150 mM thiourea also did not mitigate the effect of valinomycin and in fact enhanced its activity (Fig. 4b). In contrast to this, the presence of 10 mM tiron, a superoxide scavenger, inhibited the killing by valinomycin (Fig. 4b). In addition, inactivation of KatA, the main catalase of *B. subtilis*[40], had no effect on the killing efficiency of valinomycin (Fig. 4c), whereas deleting *sodA*, encoding the main superoxide dismutase, strongly reduced the viable count upon valinomycin treatment (Fig. 4c), and adding tiron increased the viable count of this mutant strain (Fig. S7). The Δ*sodA* mutant grew normally, indicating that the sensitivity for valinomycin is not related to a reduced cell growth (Fig. S5). Δ*sodA* cells showed almost a doubling in fluorescence of the ROS probe H2DCFDA compared to wild type cells (Fig. 4d), and the same result was found with Oxyburst Green (Fig. S6). On the other hand, the catalase mutant Δ*katA* showed comparable Oxyburst Green fluorescence as the wild type strain (Fig. S6). Finally, we tested the superoxide-specific fluorescence probe Mitosox Red[41–43], and also found a strong increase in fluorescence, which was enhanced in the Δ*sodA* mutant (Fig. S6). Apparently, depolarization of dormant *B. subtilis* cells triggers the production of lethal levels of superoxide radicals, without involvement of the Fenton reaction, which is different from the ROS generation caused by antibiotics like norfloxacin, vancomycin, and kanamycin. It is conceivable that superoxide radicals impact the barrier function of the cell membrane. However, high concentrations of paraquat, up to 4x the concentration normally used, did not affect the membrane potential (Fig. S8), suggesting that there is no apparent synergistic effect of superoxide radicals and valinomycin on the integrity of the cell membrane.

Of note, since directly damaging DNA by superoxide has not been demonstrated and since superoxide is unstable and quickly dismutates to hydrogen peroxide and then hydroxyl radical by Fenton reaction[44], the valinomycin-mediated DNA damage is more likely caused by hydrogen peroxide or hydroxyl radical, although a direct damage by superoxide cannot be ruled out.

It has been argued that the effect of ROS continuous after antibiotic treatment, during the recuperation of cells in fresh growth medium, since the presence of bipyridyl, thiourea and catalase in agar plates partially restores the viable count of *E. coli* cultures exposed to antibiotics like nalidixic acid, trimethoprim and ampicillin[45]. This post-stress killing was attributed to a self-sustained continuation in ROS production, primarily hydroxyl radicals, when a ROS threshold is exceeded[45]. To examine whether such ROS-dependent post-treatment killing occurs after valinomycin treatment, we diluted wild type and Δ*sodA* cells after 10 h of valinomycin treatment and spread them onto

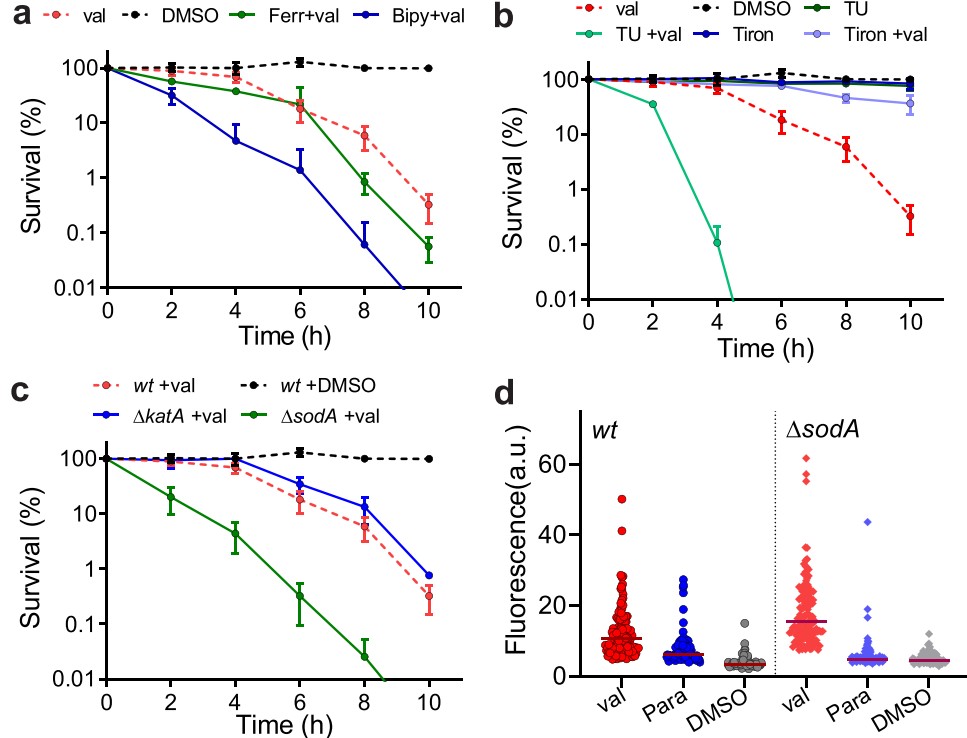

**Fig. 4 | Determining whether hydroxyl and/or superoxide radicals are formed. a** Survival curves of stationary phase cells incubated with 100 μM valinomycin (val) in the presence of either the iron chelators ferrozine (Fer, 500 μM) or 2,2'-Bipyridyl (Bip, 500 μM). Data reflect mean ± SD of three biological replicates. **b** Survival curves of stationary phase cells incubated with 100 μM valinomycin (val) in the presence of either the hydroxyl radical scavenger thiourea (TU, 150 mM) or the superoxide scavenger tiron (Tiron, 10 mM). Data reflect mean ± SD of three biological replicates. **c** Survival curves of stationary phase Δ*katA* and Δ*sodA* cells incubated with 100 μM valinomycin (val). *katA* and *sodA* code for the main catalase and superoxide dismutase, respectively. The viable counts of cells in the presence of 1 % DMSO were similar to that of wild type cells and are not indicated. Data shown reflect mean ± SD of three biological replicates. **d** ROS production in stationary phase *wt* and Δ*sod*A cells after 4 h incubation with 100 μM valinomycin (val) measured by the fluorescent ROS probe H2DCFDA. Control cells were treated with either 1 mM paraquat or 1 % DMSO for 4 h. The fluorescence intensities of 120 cells were measured microscopically and plotted.

nutrient agar plates containing either bipyridyl, thiourea, purified catalase or DMSO[45]. The latter possesses free radical-scavenging properties at high concentrations[46]. In agar plates, we could not find a concentration of bipyridyl that increased the viable count, and even 0.3 μM bipyridyl reduced the viability. Agar plates containing 5 % DMSO did not increase survival. High concentrations of thiourea appeared to be lethal to cells that have been incubated with valinomycin (not shown), which is in line with the effects of these compounds observed during valinomycin treatment (Fig. 4a, b), and a checkerboard assay showed a clear synergy for the thiourea-valinomycin combination (Table S5). Interestingly, the checkerboard data also showed a strong synergy when thiourea was combined with paraquat (Table S5). Nonetheless, low concentrations of thiourea (0.5 mM) in the agar plates slightly increased the survival rate (Fig. S9). The presence of catalase resulted in a comparable minimal increase in the viable counts (Fig. S9). However, these effect were very modest compared to what has been observed for trimethoprim-treated *E. coli* cells[45]. Finally, we also tested the presence of 10 mM tiron in plates, however, the continuous presence of this compound did not increase but instead slightly lowered the viable count (Fig. S9, Table S5). These results suggest that the killing by valinomycin cannot be attributed to post-stress induction of ROS during the recuperation phase of cells, which is in line with the clear DNA damage observed during valinomycin treatment (Fig. 2c, d).

## Possible source of superoxide production
The accumulation of lethal levels of ROS upon exposure to bactericidal antibiotics is believed to be triggered by a surge in NADH consumption that induces a burst in superoxide generation via the respiratory chain. These superoxide radicals then destabilize iron-sulfur clusters, leading to free ferrous iron, which enables the Fenton reaction, thus creating lethal levels of hydroxyl radicals[1–3]. The TCA cycle is the main source of NADH (Fig. 5a), and it was shown that inactivation of either isocitrate dehydrogenase or aconitase reduces the killing activity of norfloxacin, vancomycin and kanamycin[1]. When we inactivated *B. subtilis* pyruvate dehydrogenase, which fuels the TCA cycle, cells became not less but more sensitive to valinomycin (Fig. 5b, Δ*pdhB*), suggesting that there is no surge in NADH levels upon membrane depolarization. The other well-known source of ROS is the electron transport chain[47–50]. The *B. subtilis* electron transport chain is composed of NADH dehydrogenase, succinate dehydrogenase (complex II), cytochrome *bc* complex (complex III), and cytochrome *c* oxidase (complex IV), as schematically shown in Fig. 5a[51]. NADH dehydrogenase and complex II feed electrons to the menaquinone pool. Inactivating one of them, by either deleting *ndh*, *ndhF*, or *sdhC*, did not mitigate but rather enhanced the killing by valinomycin (Fig. 5b). Inactivation of glycerol-3-phosphate dehydrogenase, which reduces glycerol-3-phosphate to dihydroxyacetone phosphate using menaquinone[52], had no effect on the viability count (Fig. 5b, Δ*glpD*).

The absence of either the cytochrome *b* subunit (Δ*qcrB*) or the cytochrome *b/c* subunit (Δ*qcrC*) of complex III also strongly reduced the survival of valinomycin-treated cells (Fig. 5c). Interestingly, when we deleted *qcrA*, encoding the Rieske-type iron-sulfur subunit of complex III, cells became more resilient to valinomycin, and after 10 h the viable count was more than 20-fold higher than that of wild type cells (Fig. 5c).

Impairing the expression of one of the cytochrome *c* oxidase subunits of complex IV, by deleting either *ctaC*, *ctaD*, *ctaE*, or *ctaF*, made dormant cells slightly more sensitive for membrane

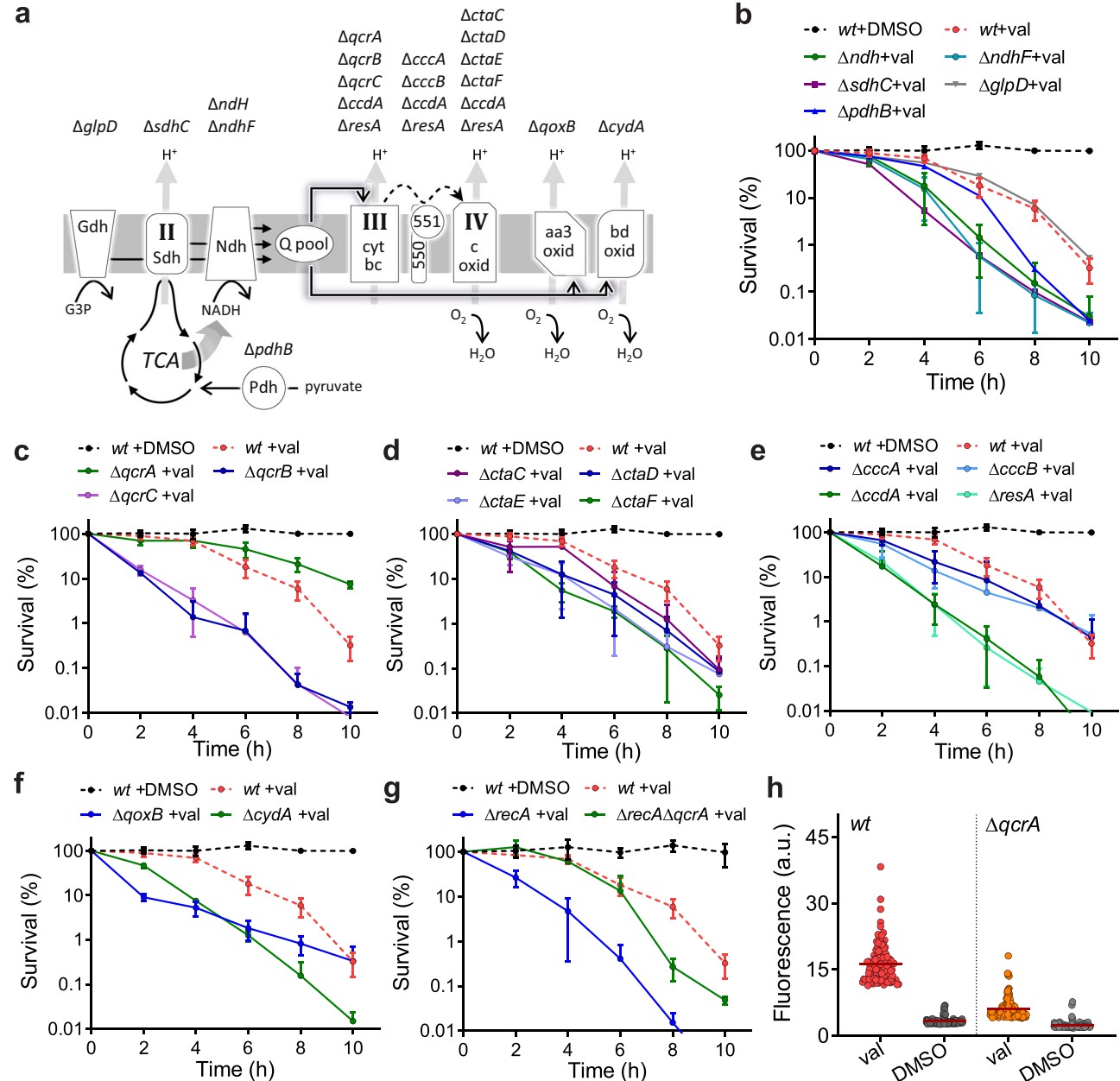

**Fig. 5 | A *qcrA* mutant increases valinomycin tolerance by generating less ROS.** **a** Schematic depiction of the key enzymes in the *B. subtilis* TCA cycle and electron transport chain. Deleted genes encoding the different components are shown above the related subunits. The different components shown are pyruvate dehydrogenase (Pdh), glycerol-3-phosphate (G3P) dehydrogenase (Gdh), succinate dehydrogenase (Complex II, Sdh), NADH dehydrogenase (Ndh), menaquinol pool (Q pool), cytochrome *bc1* complex (Complex III, cyt *bc*), cytochrome *c550* and *c551*, cytochrome-*c* oxidase (Complex IV, *c* oxid) and cytochrome *aa3* quinol oxidase (*aa3* oxid) and cytochrome *bd* ubiquinol oxidase (*bd* oxid). QcrA is the Rieske factor, menaquinol:cytochrome *c* oxidoreductase (iron-sulfur subunit), component of the cytochrome *bc1* complex. **b–g** Survival curves of the different deletion mutants grown to stationary phase and subsequently incubated with 100 μM valinomycin (val). The affected redox step in the mutants is indicated in (**a**). See main text for more details. The viable counts of the mutants treated with 1 % DMSO were similar to wild type cells. Data shown reflect mean ± SD of three biological replicates. **h** ROS production in stationary phase *wt* and Δ*QcrA* cells after 4 h incubation with 100 μM valinomycin (val) measured by the fluorescent ROS probe H2DCFDA. Control cells were treated with 1 % DMSO for 4 h. The fluorescence intensities of 120 cells were measured microscopically and plotted.

depolarization (Fig. 5d). *B. subtilis* contains two cytochrome *c* electron carriers, cytochrome *c550*, which contains a transmembrane anchor, and cytochrome *c551* that is anchored to the cell membrane via a diacyl glycerol tail (Fig. 5a)[53]. Deleting one of them did not mitigate the killing by valinomycin (Fig. 5e, Δ*cccA* or Δ*cccB*). Inactivating all the cytochrome *c*-containing components QcrBC, *c550*, *c551*, and CtaC, by removing enzymes involved in their biogenesis (Δ*ccdA*, Δ*resA*)[54,55], strongly increased the sensitivity to valinomycin (Fig. 5e). *B. subtilis* possesses two alternative cytochrome oxidases that use quinol for

their oxidation reaction, cytochrome *bd* oxidase and cytochrome *aa3* oxidase (Fig. 5a). Inactivating one of them, by deleting either *qoxB* or *cydA*, considerably reduced the viability of dormant cells when incubated with valinomycin (Fig. 5f). Of note, all mutants showed normal growth and cell densities under the growth conditions used in this study (Fig. S5). Finally, we tested whether reduction of menaquinone would reduce the sensitivity for valinomycin. Menaquinone is essential in *B. subtilis*[56], and we employed CRISPRi to conditionally reduce expression of the O-succinylbenzoate synthase MenC, which is

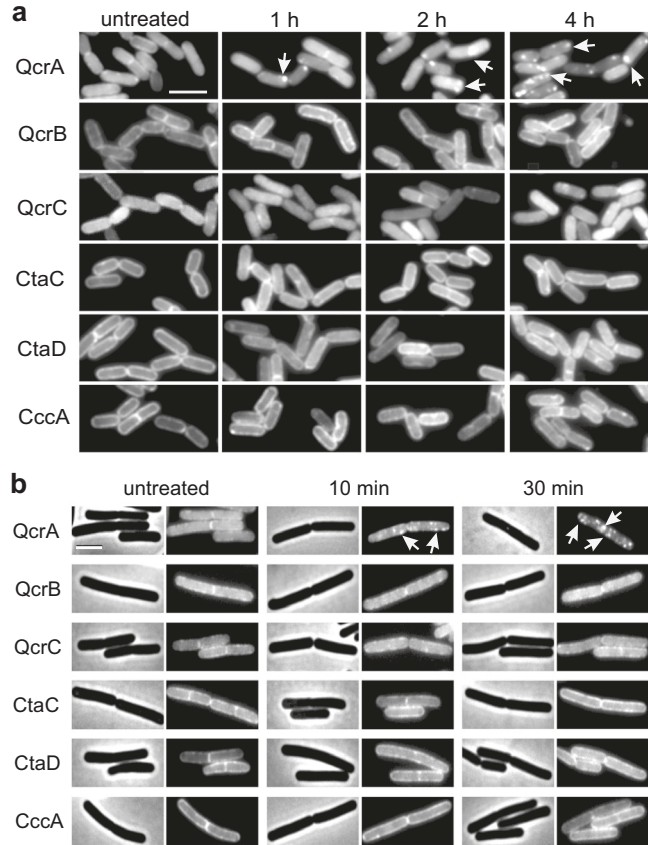

**Fig. 6 | Localization of cytochrome *bc1* and cytochrome-*c* oxidase subunits and cytochrome *cSSO* fused to GFP. a** Cellular localization of GFP tagged cytochrome *bc1* subunits QcrA (Rieske protein), QcrB and QcrC, cytochrome-*c* oxidase subunits CtaC and CtaD, and cytochrome *c550* in stationary phase cells incubated with 100 μM valinomycin for 1, 2 and 4 h. The experiment was repeated three times with similar results. **b** Same as **a** except that the cells were sampled from the exponential phase and treated with 100 μM valinomycin for only 10 to 30 min. Arrows indicate clustering of the GFP-QcrA signal. Scale bars represent 3 μm. Large field images with several cells are shown in Fig. S13 and S14. The experiment was repeated three times with similar results.

required for the synthesis of menaquinone[57]. However, depletion of *menC* also increased the sensitivity of cells for valinomycin (Fig. S10).

These surprising results suggest that not an active but rather an intact electron transport chain is important for the survival of dormant *B. subtilis* cells upon membrane depolarization, and that QcrA is a source of superoxide radicals. This Rieske protein contains a unique 2Fe-2S cluster, in which one of the two iron atoms is held in place by two histidines rather than two cysteines. This cluster accepts an electron from the quinol anion and transfers it to the cytochrome heme iron[17,58], and in fact this step is a well-known source of superoxide radicals in mitochondria[59]. To provide further support that QcrA is a likely source of the observed ROS when the membrane potential is dissipated, we deleted *qcrA* in the Δ*recA* background strain, which has an impaired DNA repair system and has been shown to be especially sensitive for valinomycin (Fig. 2). As shown in Fig. 5g, the absence of QcrA clearly attenuated the valinomycin susceptibility of dormant Δ*recA* cells. Moreover, deleting *qcrA* also mitigated the effect of valinomycin in a strain lacking both *qcrB* and *qcrC* (Fig. S11). Finally, we directly measured ROS production in the Δ*qcrA* mutant using the fluorescent ROS probes H2DCFDA (Fig. 5h), Oxyburst green and the superoxide-specific probe Mitosox Red (Fig. S12). Indeed, in the absence of the Rieske protein the average fluorescence signals after valinomycin treatment were considerably lower.

## Cellular distribution of QcrA

It is difficult to understand how QcrA can produce lethal levels of superoxide radicals upon membrane depolarization. As mentioned in the introduction, depolarization of the cell membrane leads to the delocalization of different membrane proteins, including some transmembrane proteins[14,60]. Possibly, the localization of electron transport chain components is also affected. To examine this, we constructed GFP fusions of the transmembrane proteins QcrA, QcrB, and QcrC of complex III, CtaC and CtaD of complex IV, and the main cytochrome *c* (CccA), and expressed these fusions from an ectopic locus in the genome. Of note, we have not checked whether the GFP fusions influenced their enzymatic activities, since we were only interested in their localization. As shown in Fig. 6a, all dormant cells showed a more or less uniform fluorescent membrane stain. Interestingly, incubation with valinomycin caused a strong clustering of GFP-QcrA over time, whereas the localization of the other fusions was unaffected. When we repeated the experiment with exponentially growing cells, the clustering of GFP-QcrA was already visible within 10 min after the addition of valinomycin, whereas the other fusions showed no delocalization, also not after 30 min (Fig. 6b). Possibly, the distinct clustering of QcrA upon membrane depolarization is responsible for the production of superoxide radicals.

## Discussion

In this study, we provide evidence that QcrA, the Rieske protein subunit of complex III, produces lethal levels of superoxide radicals upon depolarization of the membrane in dormant *B. subtilis* cells. Depolarization changes the distribution of QcrA in the cell membrane from smooth to spotty, which is not the case for the other two subunits QcrB and QcrC. This seems to indicate a detachment of QcrA from the other subunits of complex III. We speculate that this hampers electron transfer from the iron-sulfur cluster of QcrA to the heme of QcrB, and/or exposes the iron-sulfur cluster to molecular oxygen, thereby facilitating superoxide radical formation.

How depolarization of the membrane would lead to the disintegration of complex III is unclear. However, in previous work we have shown that the membrane potential is important for the regular distribution of different membrane proteins in *B. subtilis*, likely related to the distribution of lipids with short, unsaturated or branched fatty acids that stimulate a more liquid-disordered phase in the membrane[60]. A comparable phenomenon has been observed in yeast where the distribution of specific transmembrane proton symporters is disturbed when the membrane potential is neutralized[61], and also in mammalian cells, where membrane depolarization affects the clustering of the GTPase Ras-K[62]. These effects have been attributed to membrane potential-dependent clustering of specific lipids[61–63]. Since a lipid membrane behaves like a capacitor, a voltage difference over the membrane will apply a compression force (Maxwell pressure), which is also referred to as electrostriction. This compression is likely to affect the packing of fatty acid chains[64–66]. Possibly, the change in bilayer packing upon depolarization triggers the dissociation of QcrA from QcrBC. This dissociation might also explain why menaquinone-dependent electron transfer in the *B. subtilis* respiratory chain is facilitated by membrane energization[16].

## Mitochondrial Rieske protein

The mitochondrial Rieske iron-sulfur protein is one of the key sources of ROS in eukaryotic cells. High respiration levels increase ROS production to dangerous levels, but on the other hand, hypoxic/anoxic conditions can also trigger superoxide production by this protein[67]. The mechanism by which superoxide production is produced by hypoxia/anoxia is not clear[68], but these low oxygen conditions can lead to a reduction of the mitochondrial membrane potential[69,70]. Interestingly, the bacterium *Paracoccus denitrificans*, which contains a respiratory chain similar to that of eukaryotic mitochondria, also

shows an increase in ROS production in response to hypoxic conditions[71]. Based on these and our data, it might be interesting to examine whether the mitochondrial complex III also dissociates upon membrane depolarization, thereby exposing the Rieske subunit.

## Role of the electron transport chain

It is assumed that the production of superoxide by the Riekse subunit occurs when the oxidized iron-sulfur cluster withdraws one electron from reduced menaquinone (menaquinol)[72]. Menaquinone is reduced by NADH dehydrogenase, succinate dehydrogenase (complex II), and glycerol-3-phosphate dehydrogenase. However, inactivation of NADH dehydrogenase and complex II, and depletion of menaquinone increased the sensitivity of cells for valinomycin, as did most of the ETC mutants, with $\Delta qcrA$ being the exception (Fig. 5). Why these mutants are more sensitive for membrane depolarization remains speculation. Generally, inactivation of the ETC leads to a decrease in ATP and increase in NADH levels[73], and it is likely that this affects the repair capacity of cells.

## ROS and antibiotics

Whether the production of ROS by bactericidal antibiotics contributes to their effectivity has been hotly debated when this phenomenon was first proposed in 2007[1,3,74–79]. Our data support the finding that antibiotics can induce lethal levels of ROS. However, the mechanism of ROS production that we found differs from what has been previously described. Bactericidal antibiotics produce hydroxyl radicals from endogenous hydrogen peroxide using the Fenton reaction. The free ferrous iron required for this reaction is assumed to originate from iron-sulfur proteins, whose iron-sulfur clusters have been compromised by superoxide radicals generated by the electron transport chain that has become highly active due to a surge in NADH[80]. In fact, several studies have shown that superoxide dismutase is important for antibiotic tolerance[81–83]. How the surge in NADH happens is still not entirely clear. It has been shown that in *E. coli* the two-component regulatory systems CpxRA and ArcAB are required for the induction of ROS by antibiotics. CpxRA senses and responds to aggregated and misfolded proteins in the bacterial cell envelope, and the ArcAB system is involved in sensing oxygen availability and the concomitant transcriptional regulation of oxidative and fermentative catabolism[84]. Based on work with the ribosome inhibitor gentamycin it was proposed that mistranslation and misfolded membrane or periplasmic proteins activate the sensor CpxA, which, by an unknown mechanism, activates ArcA, as a result of which the expression of a large number of metabolic and respiratory genes are activated, leading to increased respiration rates and ultimately superoxide production[85]. Interestingly, the accumulation of misfolded cell envelope proteins by gentamycin also affects the *E. coli* membrane potential[1]. Whether a reduction of the membrane potential contributes to ROS formation in *E. coli* remains to be investigated. If this turns out to be the case, there is at least no Rieske protein involved since *E. coli* lacks complex III. Nevertheless, dormant *B. subtilis* mutants that lack QcrA still die upon membrane depolarization, albeit more slowly (Fig. 5c). This might still involve ROS as the inactivation of the RecA-based DNA repair system renders $\Delta qcrA$ cells more sensitive towards valinomycin (compare Fig. 5c and g). Possibly, a change in lipid bilayer packing due to membrane potential dissipation also affects the assembly and/or proper membrane embedding of other electron transport chain components. The resulting inefficient electron transfer between components, possibly combined with an increased exposure of their Fe-containing prosthetic groups to oxygen, might then stimulate the production of ROS.

## Relevance for persisters

*B. subtilis* is not a pathogen, so the question arises as to how relevant our findings are for the fight against clinically relevant antibiotic-tolerant persisters. Many pathogens, including *E. coli*, contain a non-canonical electron transport chain and lack complex III. However, depolarization of the cell membrane has multiple effects, including reduced ATP levels, delocalization of peripheral as well as transmembrane proteins, and possibly ROS produced by the non-canonical electron transport chain component. All these factors will affect the fitness of persister cells. A pathogen and well-known persister that lacks complex III is *Staphylococcus aureus*. Dormant *S. aureus* cells are considerably more resistant to valinomycin than *B. subtilis* (Fig. S15), however, when these *S. aureus* cells lacked one or both of its main superoxide dismutases they became substantially more sensitive for incubation with valinomycin (Fig. S15). These data support the notion that membrane depolarization can also result in the accumulation of lethal levels of superoxide radicals when no Rieske protein-containing complex III is present.

A notorious example of a pathogenic persister that does contain the Rieske protein subunit is *Mycobacterium tuberculosis*. It has been shown that the membrane potential is essential to maintain viability of non-growing mycobacterial cells[21]. Interestingly, it was also found that resistance of *M. tuberculosis* against membrane-targeting lipophilic quinazoline derivatives can arise by mutations in QcrA[86]. Both the lethal effect of membrane depolarization as well as the exposure to these quinazoline derivatives were attributed to ATP depletion. However, our results suggest that induction of superoxide radicals might be a more important factor contributing to the lethality of these interventions. In conclusion, membrane depolarization seems to be a promising method to kill bacterial persister cells since it leads to the production of lethal levels of ROS.

# Methods

## Maintenance and growth of strains

Nutrient agar (NA) (Oxoid) was used for routine selection and maintenance of both *B. subtilis* and *E. coli* strains. Supplements were added as required: chloramphenicol (5 μg/ml), erythromycin (1 μg/ml), kanamycin (2 μg/ml), spectinomycin (100 μg/ml), tetracycline (10 μg/ml), ampicillin (100 μg/ml), 0.2 % xylose and 1 mM isopropyl β-D-1-thiogalactopyranoside (IPTG). For liquid cultures, cells were cultured in either a Spizizen minimum medium (SMM)[87], modified potassium-rich Lysogeny Broth or Penassay broth (PAB). SMM contained 15 mM $(NH_4)_2SO_4$, 80 mM $K_2HPO_4$, 44 mM $KH_2PO_4$, 3 mM $Na_3C_6H_5O_7$, 0.5 % glucose, 6 mM $MgSO_4$, 0.2 mg/ml tryptophan, 0.02 % casamino acids and 0.00011 % $Fe-NH_4$-citrate. Membrane depolarization by the potassium ionophore valinomycin requires the presence of sufficient potassium ions in the medium. Therefore, cells were grown in a modified LB medium composed of 10 g/l tryptone, 5 g/l yeast extract, 50 mM HEPES pH 7.5, and 300 mM KCl, here referred to as 'KCl-LB' medium[25]. HEPES buffer was added to ensure a stable pH throughout the cultivation, which is important since the activity of valinomycin decreases upon acidification of the culture[26].

## Strain construction

Strains used in this study are listed in Table S1, and the plasmids and oligonucleotides used in this study are listed in Table S2 and S3, respectively. *B. subtilis* strain construction was performed according to established methods[88]. For the labeling of ETC components with GFP, both N- as well as C-terminal monomeric superfolder GFP (msfGFP) fusions were constructed, since it is possible that the GFP moiety interferes with membrane insertion. After microscopic inspection it appeared that only the N-terminal msfGFP fusions of QcrA, QcrB, QcrC, CccA, CtaC, and CtaD showed a clear membrane signal, indicating proper membrane insertion. These N-terminal msfGFP fusions were used for the fluorescence microscopy experiments. The xylose-inducible msfGFP fusions were constructed as follows. Target genes were PCR-amplified using forward and reverse primer pairs with genomic DNA of strain BSB1 as the template. The *amyE* integration

vector part with the xylose-inducible promoter and msfGFP sequences were amplified from pHJS105 using primer pairs EKP30 & EPK22 (for N-terminal fusion) and MW226 & EKP36 (for C-terminal fusion). Target genes and vector amplicons were purified and assembled using a two-fragment Gibson assembly. All plasmids were sequenced to confirm constructs. Subsequently, the plasmids were transformed into strain PG344 resulting in double-crossover integrations positioning the fusions in the *amyE* locus. Mutants were verified by PCR with primers TerS350 and TerS351 and by sequencing.

For constructing the CRISPRi strains, plasmid pJMP1[89] was transformed into *B. subtilis* 168 to integrate dCas9 under a xylose-inducible promoter in the *lacA* locus, resulting in strain FC67. The plasmid pJMP2[89], for constitutive sgRNA expression from the *amyE* locus, was modified to contain the desired guide using inverted PCR with the primers menC3_sgRNA_F and menC3_sgRNA_R (for sequences see Table S3), and circularized using the KLD enzyme mix (NEB). The spacer sequence against *menC* (CTGAAGGGTGATTGAATTCA) was designed using the CRISPR browser (https://crispr-browser.pasteur.cloud/)[90]. The resulting plasmids were transformed into the strain FC67 to obtain a fully functional CRISPRi system. All plasmids and integrations were sequenced to confirm correct construction. The guide effectiveness was tested by a spot assay in the presence of 1 % xylose. Cells were grown overnight in LB containing 1 µg/ml erythromycin, 25 µg/ml lincomycin, and 5 µg/ml chloramphenicol, then diluted 1:1000 in fresh LB for day culture and grown for ~6 h. Cell densities were adjusted to OD 1 and serially diluted (1:10) in PBS. Culture dilutions were spotted on agar plates of $S7_{50}$ medium[91], containing 1 % glycerol as the sole carbon source to force TCA cycle and respiration and 1 % xylose for dCas9 expression. Plates were incubated at 37 °C for 36 h.

### Minimal inhibitory concentration (MIC) measurement
The MIC values of *B. subtilis* strain PG344 were determined according to standard protocols[92]. Overnight *B. subtilis* PG344 cells were diluted in fresh medium and grown to exponential phase. Cells were then diluted to $1 \times 10^5$ CFU/ml in standard LB or KCl-LB supplemented with increasing concentrations of vancomycin, ciprofloxacin, kanamycin, valinomycin, or paraquat. The lowest concentration inhibiting visible growth after 18 h incubation at 37 °C was taken as the MIC value.

### Antibiotic survival testing
For antibiotic survival testing of vancomycin, ciprofloxacin, and kanamycin, cells were cultured in LB medium at 37 °C with aeration, whereas for valinomycin cells were cultured in KCl-LB medium under the same conditions. The exponential phase or overnight dormant cells were incubated with either 20 µg/ml vancomycin, 2 µg/ml ciprofloxacin, 10 µg/ml kanamycin, or 100 µM valinomycin. The antibiotic concentration used was 10x the MIC to ensure stability over the 10 h treatment period. 1 % DMSO, at 1 % final concentration, was used as a negative control. Supplements were added as required: 10 mM tiron, 150 mM thiourea, 0.5 mM ferrozine, and 0.5 mM 2,2'-bipyridyl. Cultures were incubated for 10 h at 37 °C with aeration, and samples were taken every 2 h in order to determine the CFU through serial dilutions and plating on LB agar. Plates were incubated overnight and colonies were counted the following morning.

The post-stress killing assay (Fig. S9) was performed as described in[45]. Stationary phase cells were incubated with valinomycin, and after 10 h diluted on nutrient agar plates containing either 0.5 mM bipyridyl, 70 mM thiourea, 5% DMSO, or 200 units purified catalase from *Aspergillus niger* (> 5000 U/mg). Catalase was diluted in water and 10 µl (200 units) was mixed with diluted cells before spreading on the plate.

### Membrane potential determination
The membrane potential levels of *B. subtilis* cells were determined using the voltage-sensitive dye 3,3'-dipropylthiadicarbocyanine iodide

[DiSC$_3$(5)] (Sigma-Aldrich) in a fluorescence microplate reader (Biotek Synergy)[25]. Briefly, early mid-exponential growth phase cells ($OD_{600}$ = ~0.4) or stationary growth phase cells (overnight cultures) were diluted in a medium supplemented with 50 µg/ml BSA to an $OD_{600}$ of 0.2 and then incubated with 1 µM DiSC$_3$(5) with shaking. The fluorescent baseline while the dye reached a stable accumulation in cells was recorded every 42 seconds using 651 nm excitation and 675 nm emission filters. When the baseline was stable, either 1 µM, 2 µM or 4 µM paraquat or 10 µM valinomycin was added to the cells. 1 µg/ml gramicidin was used as the positive control while 1 % DMSO was used as a negative control. Of note, we used a ten-times lower valinomycin concentration (10 µM) in this assay compared with the concentration used in other experiments since higher concentrations of valinomycin interfere with the fluorescence of DiSC$_3$(5). After the addition of antibiotics, the fluorescent signal was recorded for at least another 30 min.

To check that the 10 µM valinomycin concentration worked as well as the 100 µM concentration used in the experiments, we first treated cells with either 0, 1, 10, or 100 µM valinomycin for 60 min, and, after washing, incubated them with DiSC$_3$(5), to prevent interference of high concentrations of valinomycin with DiSC$_3$(5). The results, shown in Fig. S1, indicate that 10 and 100 µM valinomycin dissipate the membrane potential in comparable measures in exponentially growing cells, and that 100 µM works slightly better with stationary phase cells.

### Generation of anaerobic conditions
Strains were cultured in KCl-LB overnight at 37 °C with aeration. Following incubation, cells were mixed with either 100 µM valinomycin or 1 % DMSO. Cultures were then added to an anaerobic chamber and oxygen was removed using AnaeroGen 2.5 L sachets. Cultures were incubated for 10 h under anaerobic conditions. The CFU were determined prior to incubation and after the 10 h period through serial dilutions and plating on nutrient agar.

### Microscopy
Fluorescence microscopy experiments were performed using a Nikon Eclipse Ti equipped with a CFI Plan Apochromat DM 100x oil objective, an Intensilight HG 130-W lamp, a C11440-22CU Hamamatsu ORCA camera, and NIS-Elements software version 4.20.01 or a Nikon Eclipse Ti2 equipped with a CFI Plan Apochromat DM Lambda 100x oil objective (N.A. 1.45, W.D. 0.13 mm), a Lumencor Sola SE II FISH 365 light source, a Photometrics PRIME BSI camera, and NIS-Elements AR software version 5.21.03. Samples consisting of 0.3–0.5 µl of cells were spotted onto a microscope slide coated with a 1 % agarose film and sealed with a glass coverslip (VWR or Fisher Scientific) on top[93].

The level of cellular ROS production was measured by using the cell-permeant general ROS probes 2',7'-dichlorodihydrofluorescein diacetate (H2DCFDA), Oxyburst Green H2DCFDA succinimidyl ester (Oxyburst Green) or superoxide-specific probe Mitosox Red, from ThermoFisher[3,41–43]. H2DCFDA and Oxyburst Green were dissolved in DMSO and stored as 10 mM aliquots at −20 °C under dry argon or nitrogen, respectively. Mitosox Red was dissolved in DMSO to make a 5 mM stock under dry argon immediately before use. Overnight cultures grown in KCl-LB were 10-fold diluted in the same medium and grown at 37 °C for 2 h (till $OD_{600}$ around 2.5). The probe was added to a final concentration of 10 µM (H2DCFDA and Oxyburst Green) or 5 µM (Mitosox Red), protected from light and incubated for another 2 h until cultures entered the non-growing state ($OD_{600}$ reached and maintained around 4.0). Cells were then split and treated with 10 or 100 µM valinomycin, 1 % DMSO as negative and 1 mM paraquat (Sigma) as positive control. For visualization of fluorescence, cells of different treated timepoints were quickly washed once, resuspended in PBS, and imaged immediately. Images were analyzed using the ImageJ plugins Coli Inspector and MicrobeJ (National Institutes of Health)[94–96]. The MicrobeJ parameters for bacterial detection in phase contrast and

fluorescence intensity were set to smooth segmentation with an area of 1 μm2-max, width of 0–1.51 μm, and circularity 0–0.8. All other parameters remained at default settings.

For visualization of electron transport chain proteins, overnight cultures of GFP fusion-expressing cells or corresponding cultures grown into mid-exponential phase in KCl-LB with 0.2 % xylose at 30 °C were split and treated with 100 μM valinomycin or 1 % DMSO. Samples were taken at different time points and imaged immediately. Images were analyzed using ImageJ[94].

## Comet assay

For visualization of DNA damage, the neutral comet assay was performed according to[97,98] with modifications. Briefly, frosted microscope slides were pre-coated by dipping in 1 % agarose solution and air-drying overnight. Cells were treated with different antibiotics for 2 h and 10 μl cells were mixed with 500 μl 0.7 % low melting point agarose supplemented with 5 μg/ml RNase A (Sigma) and 0.5 mg/ml lysozyme (Sigma) at 37 °C. Immediately, 50 μl of the mixture was pipetted on a layer of low melting point agarose gel on the pre-coated slide and flattened with a coverslip, then refrigerated for 10 min at 4 °C and incubated for 30 min at 37 °C. The coverslip was removed carefully and another layer of 150 μl 0.7 % low melting point agarose was added to seal the surface. The slides were put in a tank filled with lysis solution (2.5 M NaCl, 0.1 M EDTA, 10 mM Tris base, 0.01 % SDS and 1 % Triton X-100) for 1 h at 4 °C, then washed three times by immersing into a neutralization buffer (400 mM Tris, pH 7.5) for 5 min. Following the wash, the slides were equilibrated in the electrophoresis tank with electrophoresis buffer (300 mM sodium acetate and 100 mM Tris, pH 9) for 20 min at 4 °C, then electrophoresed for 50 min at 12 V at 4 °C. After electrophoresis, the slides were wash once with neutralization buffer and dehydrated in absolute ethanol for 30 min, 70 % ethanol for 30 min, and air-dried for overnight. The dehydrated slides were stained with SYBR™ Gold (Invitrogen) and the DNA was imaged with X600 magnification using GFP filter settings. Images were analyzed using ImageJ.

## Checkerboard assay

Checkerboard assays were performed with *B. subtilis* PG344 according to[99]. The fractional inhibitory concentration index was calculated according to the formula FICI = (MICAcombi/MICAalone) + (MICBcombi/MICBalone). FICI values of ≤0.5 were defined as synergy, >0.5 to ≤4.0 as additive (no interaction), and >4.0 was defined as antagonism. Checkerboard assays were performed in biological triplicates.

## Reporting summary

Further information on research design is available in the Nature Portfolio Reporting Summary linked to this article.

# Data availability

All data supporting the findings of this study are presented in the paper, Supplementary Information or Source Data file. All strains are available upon request. Source data are provided with this paper.

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

## Acknowledgements

We would like to thank Dr. Yoshikazu Kawai and Dr. Kevin Waldron for strains. Funding for this research was provided by a Biotechnology and Biological Sciences Research Council (BBSRC) DTP Studentship BB/J014516/1 (LH), BBSRC grants BB/I004238/1 and BB/I01327X/1 (LH), NWO STW-Vici 12128 and TTW 17833 grants (LH), ZonMw grant 09120011910033 (LH), China Scholarship Council fellowship (BW), a Starting Grant from the Swedish Research Council VR, 2019-04521 (MW), and Max Planck Society grant (KT).

## Author contributions
D.G. and B.W. performed the main experiments and wrote the draft manuscript. D.G., B.W., H.S., P.G., M.W., P.G., and K.T. designed experiments and helped with data interpretation and supervision. M.S., J.W., and M.W. helped with fluorescence microscopy and spectro-scopy assays. R.R. and M.W. helped perform the checkerboard assay. F.C. and K.T. helped with CRISPRi knockdown construction. L.H. con-ceived and supervised the project.

## Competing interests
The authors declare no competing interests.
