## [Peer Review File · Nature Communications]

Membrane depolarization kills dormant *Bacillus subtilis* cells by generating a lethal dose of ROSREVIEWER COMMENTS

Reviewer #1 (Remarks to the Author):

In this study, Gray et al. use stationary phase, non-sporulating *Bacillus subtilis* to study mechanisms of valinomycin killing. Guided by knockout mutants in DNA repair systems (*recA*) and ROS-detoxification (*sodA*, *spx*), the authors examine ROS-production caused by valinomycin-induced membrane depolarization. In line with previous studies, the authors explore various iron chelators and ROS detoxifiers as well as ROS-specific probes. The authors use a panel of knockout mutants, combined with fluorescence microscopy, to link delocalization of QcrA with ROS-production and valinomycin lethality. From this work, the authors propose antibiotics can, as part of their lethality, stimulate ROS production through membrane depolarization and protein delocalization, adding to and complementing prior work in the field focused on ROS produced from metabolic stimulation and hyperactivity.

As the authors note, this work indeed supports a unique mechanism of antibiotic-induced ROS production, thus making this work an important contribution to our understanding of antibiotic mechanism of action. Further, this work is relevant to clinically-important bacterial persisters, as targeting membrane potential is an attractive target for metabolically inactive, dormant cells. The authors provide multiple forms of evidence supporting their ROS-mediated killing hypothesis (ROS-probes, knockout mutants, superoxide scavenger, fluorescence microscopy for membrane protein localization), and draw support from the well-studied mitochondrial ETC. However, some of these lines of evidence should be strengthened. As outlined in major comment 1, conclusions drawn from knockout mutants could be indirectly due to deletion of the gene of interest, complicating analysis. Here, additional characterization of population density-dependent killing and valinomycin concentration-dependent killing – for select cases to avoid unwieldy and never-ending work – would be beneficial. Additionally, the data with ROS-sensitive dye H2DCFDA is confusing, as outlined in major comment 2 below, and would benefit from addition similar probes and/or controls.

Major Comments:

1. Density-dependence and claims of cell dormancy.

a. It appears antibiotics including valinomycin were only tested at one concentration, 10x MIC of exponential phase cells. For the type of work being completed, this creates a challenge. Are differences in killing efficacy due to differences in cell physiology (due to exponential vs. stationary phase, or deletion of a gene of interest) or cell density?

i. Killing by colistin, also a membrane-targeting antibiotic with a different mechanism, is dependent on cell density (Zheng et al. – 2020 – Cell Chemical Biology)

ii. Numerous antibiotics are more or less effective at different cell densities, in part due to differences in pH at these densities (Karslake et al. – 2016 – PLoS Computational Biology)

iii. Valinomycin efficacy is dependent upon pH (“Antimicrobial activity against Gram-positive bacteria was highest at alkaline pH values...” Tempelaars et al. – 2011 – Applied and Environmental Microbiology)

b. For example, this raises the following question: As most of the mutants in Figure 5 appear more sensitive to valinomycin, could this be due to growth defects in these strains with altered ETC which could lower stationary phase carrying capacity?

c. Similarly, in Figures 1 & 6, we see that the killing kinetics of exponential and stationary phase cells differ, and Figure 1 also shows an approximately 1 log difference in T=0 density, used to calculate percent survival. Is the difference, then, due to dormancy of stationary phase cells, or differences in density?

d. Without more exploration of density-dependent effects and antibiotic concentration-dependent measurements, the status of stationary phase *B. subtilis* as “dormant persisters” is inconclusive. Yes, stationary phase is certainly a standard model system for reduced antibiotic efficacy, but without presenting more of the space around your experimental conditions, open questions can leave the reader to wonder how rigorous this finding is or if it is sensitive to underexplored conditions. While this doesn't change the direct study of valinomycin killing mechanism, it affects how the work should be discussed and presented.

2. Appropriate use of ROS-detecting dyes and conclusions of superoxide generation

a. The use of H2DCFDA is cited to Dwyer et al. – 2014 – PNAS. In Dwyer, the authors tested a panel of ROS-sensitive dyes to determine which reactive species each dye interacted with. H2DCFDA is not marked as being specific for superoxide radicals, making its use here questionable. Perhaps Oxyburst Green would be more appropriate as an additional measurement.

b. Further, it is odd that paraquat, the standard for superoxide generation, which is presented as the key ROS molecule in this story, has lower H2DCFDA induction than valinomycin (Figure 3c). What does it say about valinomycin's mechanism of action that it has a stronger signal than the positive control? Given questionable specificity of H2DCFDA for superoxide, does this suggest valinomycin primarily produces a different kind of ROS? Deletion of *sodA* leads to INCREASED valinomycin-induced H2DCFDA fluorescence, but lowers paraquat-induced values (Figure 4d). Collectively, the combination of paraquat and H2DCFDA seems inappropriate and confuses the story.

c. The authors may consider adding measurements with additional ROS-sensitive dyes for a fuller picture and/or adding exogenous H₂O₂ as an additional positive control.

Minor Comments:

- Line 53: Typo, add “s” to “bacterial peripheral membrane proteins”
- Lines 146-147: Line incorrectly states the *spx* mutant is more sensitive to valinomycin than the *recA* mutant, which is the opposite of what is shown in Fig. 3b.
- Sensitivity of a *recA* deletion is not proof of DNA damage as a mechanism of action.
 - o See Major Comment 1 above. Could the increased sensitivity of a *recA* mutant be due to reduced stationary phase carrying capacity?
 - o The *recA* finding is consistent with the ROS storyline, but is not on its own proof of ROS-induced DNA damage, thus this is only a minor comment we encourage the authors to think about. Could the *recA* mutant just be more sensitive to perturbations, even if they don't directly lead to DNA damage?

Reviewer #2 (Remarks to the Author):

The authors of this study make a case for bactericidal antibiotics killing by producing ROS. This paper revisits a highly controversial area that began with several papers from the

Collins lab that initially made this claim based on studies with *E. coli* (refs 1-3). That claim was challenged in several publications, 3 of them in *Science*, that questioned a number of results in the Collins papers, and presented one particularly telling experiment: killing under aerobic conditions was similar to anaerobic conditions (surprisingly, the authors of the present study do not mention these publications). For the majority of microbiologists, this really settled the question. The antibiotics-ROS connection became a rather fringe theory. This is not to say that the final word on this controversy has been necessarily pronounced. However, reopening this debate would require substantial and unambiguous data that adds to the subject. This is where the current study comes short. The authors study *B. subtilis* as a model, not a good choice for this line of inquiry. *B. subtilis* does not grow well under anaerobic conditions. The authors, instead of culturing *B. subtilis* under conditions where it can grow anaerobically, which would allow them to test killing by antibiotics, place aerobically cultured cells under anaerobic conditions, whereupon cells start dying. Next, they examine additional killing of these already dying cells by antibiotics. This is not an adequate experimental set up. Given the highly controversial subject, no amount of indirect data, presented in this paper, will have an impact in the absence of a clear control experiment performed under anaerobic conditions.

Reviewer #3 (Remarks to the Author):

The manuscript by Gray et al describes the novel finding that chemically induced membrane depolarization kills *Bacillus subtilis* persister cells by triggering superoxide production that is mediated by Rieske factor QcrA, the iron-sulfur subunit of respiratory chain complex III. The work is likely to be of interest to a broad readership because 1) it revealed that persister cells, thought to have low or no respiration (no source of ROS), can be killed via endogenously produced ROS, 2) it pinpointed a specific protein, QcrA, as the source of ROS, and 3) it identified superoxide, rather than hydroxyl radical or hydrogen peroxide, species commonly reported for involvement of killing with many lethal agents, as the major reactive species for killing persister cells. However, the quality of the work needs to be improved, and I have several suggestions.

Major concerns:

1. The killing and the ROS accumulation assays are all done during valinomycin treatment. Since stress-induced, ROS-mediated killing can occur after removal of the inducing stressor (in this case valinomycin), the authors need to read the Hong et al PNAS 2019 paper and then perform experiments accordingly to determine whether valinomycin-induced ROS accumulation and cell death take place after valinomycin treatment stops. That is the expected result if the authors ideas are correct. If killing occurs after valinomycin removal, then tiron, thiourea, bipyridyl, DMSO etc. should be tested as ROS mitigating agents on agar to identify the ROS that is mainly responsible for post-stress death. This set of experiments is important, because persisters are thought to have low or no respiration. It is very likely that valinomycin triggers membrane damage during drug exposure, but most ROS

production and the execution of death occur after valinomycin removal, because increased respiration, ROS accumulation, and cell death can occur when cells try to repair valinomycin-induced cellular damage during the recovery and colony forming growth period on drug-free agar.

2. Stationary-phase cells have a complex physiology, and thus they may not serve as a good example of persister populations. The authors should consider washing exponential-phase and stationary-phase cells with saline followed by incubation in saline for several hours before treating with valinomycin as an alternate way to generate and treat persister cells. If similar conclusions are reached, then it becomes more likely that what the authors observed actually reflects persister behavior.

3. Attributing persistence to superoxide and identifying its molecular origin are crucial contributions of the present work, but the supporting data are inadequate. For example, H2DCFDA is not a specific dye for superoxide -- it mainly measures hydrogen peroxide because the half-life of superoxide and hydroxyl radical are much shorter than peroxide. Experiments with MitoSox Red should now be conducted to more specifically measure superoxide. In addition, more genetic and biochemical experiments are needed to tie superoxide to killing and to gain more insight into how QcrA generates superoxide during membrane stress. Among the expectations that should be tested are 1) tiron should suppress killing of a sod mutant and of wild-type cells by valinomycin to a similar extent; 2) point mutations that affect QcrA activity (e.g. active site mutation (iron-sulfur cluster site mutation)) but not its polarization triggered by membrane potential change should help attribute superoxide generation to electron transfer rather than to non-specific protein polarization per se; 3) mutations that weaken or abolish QcrA-QcrB/C interactions, but not QcrA electron transfer activity, should become hypersensitive to valinomycin-mediated killing; 4) the qcrA-qcrB/C double mutant should behave like the qcrA single mutant and overcome qcrB/C-mediated valinomycin hypersensitivity; and 5) mutants defective in menaquinol synthesis should reduce valinomycin-mediated killing.

4. The literature contains reports that attribute persistence to superoxide. These should be cited to establish author credibility.

5. Superoxide itself may damage membrane and affect membrane potential, which makes superoxide and valinomycin synergistically exhibiting synthetic lethal effect. Such a possibility needs to be addressed since that would inflate the role of superoxide over that of other reactive oxygen species in contribution to killing. For example, a recA mutant is hypersensitive to valinomycin, indicating that valinomycin leads to DNA damage. If valinomycin mainly produces superoxide, then it is difficult to explain how DNA is damaged by superoxide since hydroxyl radical, not superoxide, is the major ROS that causes DNA damage. Thus, a cascade of superoxide-hydrogen peroxide-hydroxyl radical rather than superoxide alone must be involved in valinomycin-mediated killing.

Minor issues

Lines 38-41: DNA synthesis inhibitors are the strongest ROS inducers and thus should be added here.

Lines 41-42: No evidence for this statement although the lethal cellular damage might be largely due to hydroxyl radical.

Lines 328-329: need reference citations for this statement

Line 418: change was to were

Lines 448-451: This might be problematic because 10-fold lower concentrations of valinomycin may not behave the same as full dose in terms of triggering membrane potential change or ROS, and surely not for killing (the low concentration used here may not kill at all and thus would not reflect the true situation). You should consider using a full dose range of valinomycin and then wash it out before using the dye for measuring membrane potential.

Figure 2, a and b (maybe c) should be moved to supplementary data, as these are all negative results.

Figure 5e legend: Is there a typo in *cccdA*?

Figure 6: discrete dots, although less dramatic than seen with *qcrA*, were also evident with *qcrB* and *qcrC*. You need to explain this observation.

Point-by-point reply to reviewers' comments

We would like to thank the reviewers for their efforts and useful comments. The revision has taken a substantial amount of time due to consequences related to the final Covid lockdown. Our apologies for this.

To clearly indicate our replies, we have written them in blue. A list with references used is added to the end of this rebuttal.

Reviewer #1

In this study, Gray et al. use stationary phase, non-sporulating *Bacillus subtilis* to study mechanisms of valinomycin killing. Guided by knockout mutants in DNA repair systems (*recA*) and ROS-detoxification (*sodA*, *spx*), the authors examine ROS-production caused by valinomycin-induced membrane depolarization. In line with previous studies, the authors explore various iron chelators and ROS detoxifiers as well as ROS-specific probes. The authors use a panel of knockout mutants, combined with fluorescence microscopy, to link delocalization of QcrA with ROS-production and valinomycin lethality. From this work, the authors propose antibiotics can, as part of their lethality, stimulate ROS production through membrane depolarization and protein delocalization, adding to and complementing prior work in the field focused on ROS produced from metabolic stimulation and hyperactivity.

As the authors note, this work indeed supports a unique mechanism of antibiotic-induced ROS production, thus making this work an important contribution to our understanding of antibiotic mechanism of action. Further, this work is relevant to clinically-important bacterial persisters, as targeting membrane potential is an attractive target for metabolically inactive, dormant cells. The authors provide multiple forms of evidence supporting their ROS-mediated killing hypothesis (ROS-probes, knockout mutants, superoxide scavenger, fluorescence microscopy for membrane protein localization), and draw support from the well-studied mitochondrial ETC. However, some of these lines of evidence should be strengthened. As outlined in major comment 1, conclusions drawn from knockout mutants could be indirectly due to deletion of the gene of interest, complicating analysis. Here, additional characterization of population density-dependent killing and valinomycin concentration-dependent killing— for select cases to avoid unwieldy and never-ending work — would be beneficial. Additionally, the data with ROS-sensitive dye H2DCFDA is confusing, as outlined in major comment 2 below, and would benefit from addition similar probes and/or controls.

Major Comment 1. Density-dependence and claims of cell dormancy.

a. It appears antibiotics including valinomycin were only tested at one concentration, 10x MIC of exponential phase cells. For the type of work being completed, this creates a challenge. Are differences in killing efficacy due to differences in cell physiology (due to exponential vs. stationary phase, or deletion of a gene of interest) or cell density? i. Killing by colistin, also a

membrane-targeting antibiotic with a different mechanism, is dependent on cell density (Zheng et al. – 2020 – Cell Chemical Biology), ii. Numerous antibiotics are more or less effective at different cell densities, in part due to differences in pH at these densities (Karslake et al. – 2016 – PloS Computational Biology), iii. Valinomycin efficacy is dependent upon pH (“Antimicrobial activity against Gram-positive bacteria was highest at alkaline pH values...” Tempelaars et al. – 2011 – Applied and Environmental Microbiology)

b. For example, this raises the following question: As most of the mutants in Figure 5 appear more sensitive to valinomycin, could this be due to growth defects in these strains with altered ETC which could lower stationary phase carrying capacity?

c. Similarly, in Figures 1 & 6, we see that the killing kinetics of exponential and stationary phase cells differ, and Figure 1 also shows an approximately 1 log difference in T=0 density, used to calculate percent survival. Is the difference, then, due to dormancy of stationary phase cells, or differences in density?

d. Without more exploration of density-dependent effects and antibiotic concentration-dependent measurements, the status of stationary phase *B. subtilis* as “dormant persisters” is inconclusive. Yes, stationary phase is certainly a standard model system for reduced antibiotic efficacy, but without presenting more of the space around your experimental conditions, open questions can leave the reader to wonder how rigorous this finding is or if it is sensitive to underexplored conditions. While this doesn’t change the direct study of valinomycin killing mechanism, it affects how the work should be discussed and presented.

We thank the reviewer for pointing out that growth conditions, especially pH and cell density, can influence the efficacy of antimicrobials. These are indeed important conditions that we need to address. The sensitivity of valinomycin to pH is known to us, and is, in fact, the reason why we included 50 mM Hepes pH 7.5 in the medium. We listed the addition of Hepes buffer in the Experimental Procedure paragraph of the original manuscript (line 386), but we did not explain this in detail. We have now added this explanation and mention both in the Results section and in the Experimental Procedures that valinomycin is pH-sensitive, and that this is the reason why we added Hepes buffer to the growth medium (lines 111-113 and 459-461 in the revision). In addition, we have now confirmed that the pH in this medium remained stable even in the stationary phase after 10 h valinomycin treatment, also in cultures of key mutants used in this study (new Fig. S2). We mention this in lines 113-115 of the revised manuscript.

We are also aware that the cell density affects the activity of antibiotics, especially membrane-targeting antibiotics. This is why we always use constant cell densities in our experiments, except for the very first experiment where we initially compared the sensitivity of exponential phase cultures with stationary phase cultures (Fig. 1). However, we agree that it is a good idea to clearly address the effect of cell density in the paper. To this end, we tested the efficacy of valinomycin on 10x concentrated exponential phase cultures and 10x diluted stationary phase cultures, as shown in the new Fig. S3. This experiment showed that there is a noticeable yet not decisive effect on the activity of valinomycin related to the cell

concentration. We discuss this now in the revised main text in lines 120-127.

In addition, to check how robust the mutants grew and remain stable during the stationary phase, or in other words, whether the "stationary phase carrying capacity" of the mutants are comparable to the wild type strain, we measured the growth curves of all the different mutants that we have tested. This revealed that under our growth conditions all mutants showed comparable growth rates and reached comparable ODs, except for the $\Delta recA$ strain (new Fig. S5a). However, although the $\Delta recA$ mutant grew slower, this strain reached a normal OD during the stationary phase (new Fig. S5a, b), which we now discuss in line 148-151 in the revision.

For several key mutants used in study, including, $\Delta recA$, $\Delta sodA$, $\Delta qcrA$, and $\Delta qcrB$, we now show detailed OD and CFU measurements at the beginning and end of the valinomycin treatment in the new Fig. S5b. These data confirm that the concentration of cells of these mutants is comparable to that of the wild type culture. We refer to the growth rate data and OD measurements of the mutants shown in Fig. S5 in lines 148-151 ($\Delta recA$), 213-214 ($\Delta sodA$), 287-289 (TCA and ETC mutants, and $\Delta qcrA$, $\Delta qcrB$) in the revision.

Major Comment 2. Appropriate use of ROS-detecting dyes and conclusions of superoxide generation.

a. The use of H2DCFDA is cited to Dwyer et al. – 2014 – PNAS. In Dwyer, the authors tested a panel of ROS-sensitive dyes to determine which reactive species each dye interacted with. H2DCFDA is not marked as being specific for superoxide radicals, making its use here questionable. Perhaps Oxyburst Green would be more appropriate as an additional measurement.

b. Further, it is odd that paraquat, the standard for superoxide generation, which is presented as the key ROS molecule in this story, has lower H2DCFDA induction than valinomycin (Figure 3c). What does it say about valinomycin's mechanism of action that it has a stronger signal than the positive control? Given questionable specificity of H2DCFDA for superoxide, does this suggest valinomycin primarily produces a different kind of ROS? Deletion of *sodA* leads to INCREASED valinomycin-induced H2DCFDA fluorescence, but lowers paraquat-induced values (Figure 4d). Collectively, the combination of paraquat and H2DCFDA seems inappropriate and confuses the story.

c. The authors may consider adding measurements with additional ROS-sensitive dyes for a fuller picture and/or adding exogenous H₂O₂ as an additional positive control.

Unfortunately, the table in Fig. 1 of the Dwyer et al. paper (Dwyer et al. 2014) gives an oversimplified picture of ROS species specificity, and is in fact partially incorrect. Firstly, there is no literature indicating that Oxyburst Green can specifically and only detect superoxide radicals. Also the references on the detection of superoxide radicals by Oxyburst green provided in the 2010 Molecular Probes manual chapter 18, which might be the information

on which the table in Fig. 1 was based, do not indicate that this is the case (Ryan et al. 1990, Ohkuro et al. 1994). Secondly, Thermo Fisher, who took over Molecular Probes, is no longer advertising Oxyburst Green as a specific superoxide radical probe. Thirdly, in the Dwyer et al. paper Oxyburst Green still gives a clear fluorescence response upon treatment of ampicillin and norfloxacin, whereas the authors argue that their treatment produces primarily hydrogen peroxide, and fourthly, in an unrelated research project, we have tested how specific Oxyburst Green is in *B. subtilis* and found that hydrogen peroxide gives a higher fluorescence signal with Oxyburst Green than paraquat (Fig. S2 in (Schafer et al. 2023)), confirming that Oxyburst Green is, unfortunately, not specific for superoxide radicals.

We have used H2DCFDA, one of the most commonly used and robust ROS probes, to show that ROS is generated, and use other experiments to come to the conclusion that primarily superoxide radicals are formed, including the effect of hydroxyl radical and superoxide scavengers (thiourea and tiron, respectively) and the difference in sensitivity of superoxide and catalase mutants. This conclusion fits the finding that the Rieske protein, a well-known source of superoxide radicals, is a major player in the killing of cells upon membrane depolarization. Of note, we have tried the more superoxide-selective fluorescent probe MitoSox Red, but this probe appeared to be toxic to *B. subtilis* cells and led to rapid cell lysis. Therefore, this probe could not be used.

Despite our reservations about the specificity of Oxyburst Green, we have now also tested this probe, and the new Fig. S6 shows that the fluorescence of Oxyburst Green also increases upon incubation of cells with valinomycin, and that this increase is higher in the superoxide dismutase mutant $\Delta sodA$ mutant but not in the catalase mutant $\Delta katA$. Moreover, the increase in Oxyburst Green fluorescence was reduced in $\Delta qcrA$ cells lacking the Rieske protein (Fig. S12). This data is well in line with our original results, and we mention now the use of Oxyburst Green in the revision in lines 191-192 for wild type cells, lines 216-219 for $\Delta sodA$ and $\Delta katA$ cells, and in line 308 for $\Delta qcrA$ cells.

With regards to the remark of the reviewer, why the addition of paraquat results in weaker H2DCFDA fluorescence compared to valinomycin, we can only speculate. With the Oxyburst Green probe we see a clear increase in fluorescence with paraquat especially in the superoxidase $\Delta sodA$ mutant (Fig. S6), but also with Oxyburst Green the increase in fluorescence is higher in the presence of valinomycin (Fig. S6), suggesting that under our conditions valinomycin creates more ROS than paraquat.

Minor Comments:

- Line 53: Typo, add “s” to “bacterial peripheral membrane proteins”

Thank you for pointing this out, we have corrected the typo.

- Lines 146-147: Line incorrectly states the *spx* mutant is more sensitive to valinomycin than the *recA* mutant, which is the opposite of what is shown in Fig. 3b.

Thanks for pointing out this mix-up. The sentence has now been adjusted (line 182 in the revision).

- Sensitivity of a *recA* deletion is not proof of DNA damage as a mechanism of action.
 - o See Major Comment 1 above. Could the increased sensitivity of a *recA* mutant be due to reduced stationary phase carrying capacity?
 - o The *recA* finding is consistent with the ROS storyline, but is not on its own proof of ROS-induced DNA damage, thus this is only a minor comment we encourage the authors to think about. Could the *recA* mutant just be more sensitive to perturbations, even if they don't directly lead to DNA damage?

This is a valid point. Firstly, we have now tested how well the $\Delta recA$ mutant grows, and although this mutant shows slower growth during the exponential phase, it reaches normal cell densities in the stationary phase and does not show increased cell lysis over time (New Fig. S5). Therefore, it appears that the sensitivity of the $\Delta recA$ mutant is not related to a reduced stationary phase carrying capacity. We allude to this in lines 148-151 in the revision.

Secondly, and in line with the comment of the reviewer, it would be very informative to directly show that there is an increase in DNA damage when cells are treated with valinomycin. A common method in the eukaryotic field to directly show DNA breaks in cells is the so called "comet assay". In this assay, cells are exposed to an electric field after treatment with DNA-damaging compounds and cell lysis, and the migration of the nuclear DNA is monitored microscopically (Azqueta and Collins 2013). DNA exhibiting strand breaks will produce a longer DNA tail. We have tested whether we could use such a comet assay to examine DNA damage upon treatment with valinomycin, and this was indeed the case. The incubation with valinomycin resulted in DNA comet tails that were on average 3x longer compared to untreated cells. Since we think that this direct DNA damage assay provides conclusive information and is a useful tool to assess DNA damage in bacterial cells, we show this data now in the main text in the new Fig. 2c, d, and describe this assay in lines 151-160 in the revision.

Reviewer #2:

The authors of this study make a case for bactericidal antibiotics killing by producing ROS. This paper revisits a highly controversial area that began with several papers from the Collins lab that initially made this claim based on studies with *E. coli* (refs 1-3). That claim was challenged in several publications, 3 of them in Science, that questioned a number of results in the Collins

papers, and presented one particularly telling experiment: killing under aerobic conditions was similar to anaerobic conditions (surprisingly, the authors of the present study do not mention these publications). For the majority of microbiologists, this really settled the question. The antibiotics-ROS connection became a rather fringe theory. This is not to say that the final word on this controversy has been necessarily pronounced. However, reopening this debate would require substantial and unambiguous data that adds to the subject. This is where the current study comes short. The authors study *B. subtilis* as a model, not a good choice for this line of inquiry. *B. subtilis* does not grow well under anaerobic conditions. The authors, instead of culturing *B. subtilis* under conditions where it can grow anaerobically, which would allow them to test killing by antibiotics, place aerobically cultured cells under anaerobic conditions, whereupon cells start dying. Next, they examine additional killing of these already dying cells by antibiotics. This is not an adequate experimental set up. Given the highly controversial subject, no amount of indirect data, presented in this paper, will have an impact in the absence of a clear control experiment performed under anaerobic conditions.

We agree with the referee that the theory that ROS is instrumental in the activity of certain antibiotics is still controversial, and in fact we have mentioned this in the Discussion: "*Whether the production of ROS by bactericidal antibiotics contributes to their effectivity has been hotly debated when this phenomenon was first proposed in 2007*". We also did provide 3 references that are critical of the antibiotic-killing-by-ROS theory: (Feld et al. 2012, Imlay 2013, Paulander et al. 2014). There is no particular reason why we did not cite the 3 papers mentioned by the reviewer, and we are happy to add them (Foti et al. 2012, Keren et al. 2013, Liu and Imlay 2013). They are now referred to in the revised manuscript in line 384.

However, we do not agree with the reviewer that the majority of microbiologists consider this now a fringe theory, and this has also not been suggested by the two other reviewers. In fact, several papers have addressed and countered key issues raised in the ROS-critical papers that the reviewer is referring to, including (Dwyer et al. 2014, Zhao and Drlica 2014, Dwyer et al. 2015, Zhao et al. 2015). Moreover, there is a steady stream of new papers being published that show a role for ROS in killing by antibiotics, and that are not linked to the Collins lab, e.g. (Hong et al. 2019, Lam et al. 2020, Arce-Rodriguez et al. 2022, Shee et al. 2022, Guo et al. 2023), and e.g. a very recent publication in Nature Communications: (Kawai et al. 2023).

Maybe even more importantly, our study describes a different type of ROS species caused by a membrane potential-dissipating antibiotic, namely superoxide radicals, whereas the group of Collins and others described the production of hydroxyl radicals by antibiotics that affect translation and cell wall synthesis (Kohanski et al. 2007). Moreover, we provide a novel explanation for the generation of these superoxide radicals that has broad implications, as we describe in the Discussion. Thus, our findings fundamentally differ from previous studies on the production of hydroxyl radicals by antibiotics. Finally, we do not agree that our study is irrelevant because we cannot grow the model system anaerobically. Firstly, most bacterial

pathogens infect under aerobic conditions, and secondly, the outcome of anaerobic experiments does not provide an explanation for our crucial observations, such as the direct observed DNA damage and ROS production in cells, and the fact that $\Delta recA$, $\Delta sdpC$ and especially $\Delta sodA$ mutants are hypersensitive, and why the absence of the Rieske protein makes cells less sensitive to valinomycin.

Reviewer #3:

The manuscript by Gray et al describes the novel finding that chemically induced membrane depolarization kills *Bacillus subtilis* persister cells by triggering superoxide production that is mediated by Rieske factor QcrA, the iron-sulfur subunit of respiratory chain complex III. The work is likely to be of interest to a broad readership because 1) it revealed that persister cells, thought to have low or no respiration (no source of ROS), can be killed via endogenously produced ROS, 2) it pinpointed a specific protein, QcrA, as the source of ROS, and 3) it identified superoxide, rather than hydroxyl radical or hydrogen peroxide, species commonly reported for involvement of killing with many lethal agents, as the major reactive species for killing persister cells. However, the quality of the work needs to be improved, and I have several suggestions.

Major concerns:

1. The killing and the ROS accumulation assays are all done during valinomycin treatment. Since stress-induced, ROS-mediated killing can occur after removal of the inducing stressor (in this case valinomycin), the authors need to read the Hong et al PNAS 2019 paper and then perform experiments accordingly to determine whether valinomycin-induced ROS accumulation and cell death take place after valinomycin treatment stops. That is the expected result if the authors ideas are correct. If killing occurs after valinomycin removal, then tiron, thiourea, bipyridyl, DMSO etc. should be tested as ROS mitigating agents on agar to identify the ROS that is mainly responsible for post-stress death. This set of experiments is important, because persisters are thought to have low or no respiration. It is very likely that valinomycin triggers membrane damage during drug exposure, but most ROS production and the execution of death occur after valinomycin removal, because increased respiration, ROS accumulation, and cell death can occur when cells try to repair valinomycin-induced cellular damage during the recovery and colony forming growth period on drug-free agar.

We thank the reviewer for pointing out the (Hong et al. 2019) paper, describing the killing by ROS after antibiotic stress (post-stress killing). The reviewer mentions that we should find the same when using valinomycin ("*That is the expected result if the authors ideas are correct*"). However, we believe that this is not necessarily the case. Hong et al. describe that this post-antibiotic stress phenomenon is primarily caused by hydroxyl radicals, caused by the Fenton reaction, since in their experiments the presence of the iron chelator piperidyl, and the addition of catalase or the hydroxyl radical scavenger thiourea mitigate the post-stress killing

(Hong et al. 2019). However, in our manuscript we have shown that valinomycin creates superoxide radicals, and that this effect cannot be mitigated by either piperidyl or thiourea. Thus, the stress caused by valinomycin, i.e. dissipation of the membrane potential, seems to be fundamentally different from the stress caused by e.g. the DNA, dihydrofolate synthesis, and cell wall targeting antibiotics nalidixic acid, trimethoprim and ampicillin described in the (Hong et al. 2019) paper. Of note, our new comet assay also shows that DNA damage occurs during valinomycin treatment (new Fig. 2c-d).

Nevertheless, inspired by the remarks of the reviewer, we became curious whether there is any post-valinomycin stress killing. Of note, if this is the case, such stress seems to be at least unrelated to hydroxyl radicals, since a strain lacking the main catalase ($\Delta katA$) shows the same viable count numbers as the wild type strain (Fig. 4c). We have tested the agar recuperation assay described in the (Hong et al. 2019) paper, and tested the viable count, after valinomycin treatment, using plates containing either thiourea, DMSO, tiron, or purified catalase. Thiourea appeared to be lethal, as we had shown before in Fig. 4b, and neither the presence of tiron nor DMSO increased the recovery (new Fig. S9). However, the presence of purified catalase shows a slight increase in viable counts, but not close to the recovery described in the (Hong et al. 2019) paper. We also tested the effect of the iron chelator bipyridyl in plates, but this compound also appeared to be lethal for valinomycin-treated cells, again in line with what we have described in Fig. 4a. Therefore, it seems that the ROS-related killing induced by valinomycin does not occur post-stress, during the recuperation of cells. We have now included a discussion on the post-antibiotic ROS killing as described in the (Hong et al. 2019) paper, and our experimental assessment of this phenomenon, in lines 227-248 in the revision.

2. Stationary-phase cells have a complex physiology, and thus they may not serve as a good example of persister populations. The authors should consider washing exponential-phase and stationary-phase cells with saline followed by incubation in saline for several hours before treating with valinomycin as an alternate way to generate and treat persister cells. If similar conclusions are reached, then it becomes more likely that what the authors observed actually reflects persister behavior.

We appreciate the suggestion of the reviewer because it touches upon a recurring question what a good persister model is, for which there is, in our opinion, not a clear answer. The reason for this is that antibiotic-tolerant persister cells can emerge by diverse mechanisms, including activation of the stringent response, toxin/antitoxin systems, and other mechanisms that trigger a temporary non-growing state, such as the formation of biofilms that limits access to nutrients. The stationary phase cells that we have used in our study resemble the latter type of persister cells, characterized primarily by low metabolic activity. There are other papers that use stationary phase cells as a model for persisters, as we have mentioned in the manuscript (e.g. (Orman and Brynildsen 2015)). However, we have followed the reviewer's suggestion and examined whether washing and incubating exponentially growing cells in

buffer for 2 hours to stop growth would make them tolerant to antibiotics, including vancomycin, ciprofloxacin, and kanamycin, but this was not the case and cells remained sensitive for all three antibiotics. We mention this now in the revision in line 93-95.

3. Attributing persistence to superoxide and identifying its molecular origin are crucial contributions of the present work, but the supporting data are inadequate. For example, H2DCFDA is not a specific dye for superoxide -- it mainly measures hydrogen peroxide because the half-life of superoxide and hydroxyl radical are much shorter than peroxide. Experiments with MitoSox Red should now be conducted to more specifically measure superoxide.

We have tried MitoSox Red, but unfortunately *B. subtilis* cells lyse rapidly when incubated with this probe, making it unusable. However, we have now tested another ROS probe, Oxyburst Green (see also reply to reviewer 1), which gave comparable results as H2DCFDA (Fig. S6 and S12).

In one of the earlier papers on ROS formation by antibiotics it was postulated that Oxyburst Green was specific for superoxide radicals, Fig. 1 in (Dwyer et al. 2014), however, as we have pointed out in our reply to reviewer 1, this specificity is questionable. Here, we would like to stress again that we base our conclusion that valinomycin creates superoxide radicals on a number of different observations unrelated to the specificity of the ROS probes, including the finding that (i) the superoxide scavenger tiron suppresses the valinomycin effect, whereas the hydroxyl radical scavenger thiourea does not. (ii) The superoxide dismutase $\Delta sodA$ mutant is very sensitive to valinomycin, whereas the catalase $\Delta katA$ mutant is not. (iii) The finding that a $\Delta sodA$ mutant gives the highest fluorescence with Oxyburst Green, whereas the $\Delta katA$ mutant produces a fluorescence that is comparable to the wild type strain (Fig. S6), (iv) the fact that the iron chelators ferrozine and bipyridyl do not mitigate the killing by valinomycin, thus making hydroxyl radical formation by means of the Fenton reaction unlikely, and finally, (v) the finding that only the absence of the Rieske protein, a well-known source of superoxide radicals, attenuates the killing by valinomycin. We feel that based on these results it is reasonable to propose that valinomycin creates primarily superoxide radicals. The new Oxyburst Green data are now mentioned in the revision in lines 191-192 for wild type cells, lines 216-219 for $\Delta sodA$ and $\Delta katA$ cells, and in line 308 for $\Delta qcrA$ cells.

In addition, more genetic and biochemical experiments are needed to tie superoxide to killing and to gain more insight into how QcrA generates superoxide during membrane stress. Among the expectations that should be tested are 1) tiron should suppress killing of a sod mutant and of wild-type cells by valinomycin to a similar extent; 2) point mutations that affect QcrA activity (e.g. active site mutation (iron-sulfur cluster site mutation)) but not its polarization triggered by membrane potential change should help attribute superoxide generation to electron transfer rather than to non-specific protein polarization per se; 3) mutations that weaken or abolish QcrA-QcrB/C interactions, but not QcrA electron transfer

activity, should become hypersensitive to valinomycin-mediated killing; 4) the *qcrA-qcrB/C* double mutant should behave like the *qcrA* single mutant and overcome *qcrB/C*-mediated valinomycin hypersensitivity and 5) mutants defective in menaquinol synthesis should reduce valinomycin-mediated killing.

Suggestion 1: We have tested whether tiron can suppress the killing of a sensitive *ΔsodA* mutant strain to a similar extent as the wild type, and this was indeed the case, as shown in the new Fig. S7. We mention this experiment now in lines 212-213 in the revision. Of note, since the *ΔsodA* mutant is much more sensitive for valinomycin compared to the wild type strain, the presence of tiron does not restore the viability of the *ΔsodA* mutant to the levels seen with wild type cells.

Suggestion 2 & 3: We feel that an extensive mutagenesis study of QcrA, B, and C is beyond the scope of the current study, and something for a later study that is aimed at the mechanism of superoxide production by QcrA. Of note, such a study will be challenging since it was shown that mutations of the QcrA iron-sulfur cluster site lead to failure of QcrA insertion into the membrane (Goosens et al. 2014).

Suggestion 4: We have tested whether deletion of *qcrA* in a *ΔqcrBC* deletion mutant mitigates the killing by valinomycin, and this was indeed the case (new Fig. S11), mentioned in line 305-306 in the revision.

Suggestion 5: Unfortunately, menaquinone is essential and the related biosynthetic genes cannot be deleted in *B. subtilis*. We have tried to use an inducible menaquinone synthetase system, but were unable to sufficiently deplete the menaquinone pool, judging from growth experiments on medium with glycerol as main carbon source (data not shown). As an alternative method we applied CRISPRi, and this time we were able reduce the expression of *menC*, encoding the O-succinylbenzoate synthase required for menaquinone synthesis, sufficiently to see a clear growth defect on glycerol medium (new Fig. S10). Interestingly, a reduction in *menC* levels made cells more sensitive to valinomycin (new Fig. S10), comparable to the inactivation of many other TCA and ETC components. These new results are described in lines 289-293 in the revision. Of note, it is very likely that the downregulation of *menC* only reduces menaquinone levels, leaving enough to transfer electrons to QcrA. We have now adjusted the Discussion to include the CRISPRi results (line 375 in the revision).

4. The literature contains reports that attribute persistence to superoxide. These should be cited to establish author credibility.

We thank the reviewer for pointing this out and refer to these reports in line 394 of the Discussion.

5. Superoxide itself may damage membrane and affect membrane potential, which makes superoxide and valinomycin synergistically exhibiting synthetic lethal effect. Such a possibility needs to be addressed since that would inflate the role of superoxide over that of other reactive oxygen species in contribution to killing. For example, a *recA* mutant is hypersensitive to valinomycin, indicating that valinomycin leads to DNA damage. If valinomycin mainly produces superoxide, then it is difficult to explain how DNA is damaged by superoxide since hydroxyl radical, not superoxide, is the major ROS that causes DNA damage. Thus, a cascade of superoxide-hydrogen peroxide-hydroxyl radical rather than superoxide alone must be involved in valinomycin-mediated killing.

Here, we have difficulties following the reasoning of the reviewer. We do not see how a possible synergistic effect of superoxide and valinomycin on the membrane would inflate the role of superoxide over that of other ROS species. We also do not have any indication for the suggested ROS cascade. Firstly, a catalase mutant ($\Delta katA$) is not more sensitive for valinomycin than the wild type strain (Fig. 4c), and produces the same levels of ROS as the wild type strain, according to the Oxyburst Green ROS probe (Fig. S6). Moreover, the hydroxyl-radical quencher thiourea does not mitigate killing by valinomycin (Fig. 4b), suggesting that the involvement of peroxide-hydroxyl radicals is limited. In addition, there are also studies that show that superoxide radicals can damage DNA, see e.g. (Misiaszek et al. 2004), and finally, we do not say that valinomycin *exclusively* produces superoxide radicals, but that it is the main component. Additionally, we have now tested whether superoxide radicals formed by the addition of only paraquat would affect the membrane potential, and we could not find any effect with up to 4 times the concentration used in the fluorescent ROS probe experiments (new Fig. S8). This is now discussed in the revision in lines 222-226.

Minor issues

Lines 38-41: DNA synthesis inhibitors are the strongest ROS inducers and thus should be added here.

We thank the reviewer for pointing out this omission. We have now added this information in the revision (line 44-45).

Lines 41-42: No evidence for this statement although the lethal cellular damage might be largely due to hydroxyl radical.

We are not sure what the reviewer means by this. In case this remark is related to the production of hydroxyl radicals by the Fenton reaction, the references are given in the next sentence (line 48 in the revision).

Lines 328-329: need reference citations for this statement

We have added the correct reference (Kohanski et al. 2008).

Line 418: change was to were

Thank you for pointing this out, this has been corrected.

Lines 448-451: This might be problematic because 10-fold lower concentrations of valinomycin may not behave the same as full dose in terms of triggering membrane potential change or ROS, and surely not for killing (the low concentration used here may not kill at all and thus would not reflect the true situation). You should consider using a full dose range of valinomycin and then wash it out before using the dye for measuring membrane potential.

The valinomycin concentration that we use is well above concentrations generally used to dissipate the membrane potential, and so is the 10-fold lower concentration (= 10 μ m) used in the DiSC3(5) assay described in the paper. But we agree that it is important to show this, and we have followed the suggestion of the reviewer by first adding valinomycin and then wash it and adding DiSC3(5). The results of this experiment are shown in the new Fig. S1, and confirm that both 10 and 100 μ m valinomycin dissipate the membrane potential. This control is described in lines 544-550 in the revision.

Figure 2, a and b (maybe c) should be moved to supplementary data, as these are all negative results.

We have moved Fig. 2 a and b to the supplementary data (new Fig. S4).

Figure 5e legend: Is there a typo in *cccdA*?

No, the Δ *cccA* and Δ *ccdA* are two different mutants. *cccA* encodes cytochrome *c550* and *ccdA* encodes the membrane-embedded thiol-disulfide oxidoreductase involved in cytochrome *c* synthesis.

Figure 6: discrete dots, although less dramatic than seen with *qcrA*, were also evident with *qcrB* and *qcrC*. You need to explain this observation.

Here, we have difficulties seeing what the reviewer is seeing. It is quite common that there is some variation between cells, but when we look at the bigger picture, like the wide-field images shown in Fig. S13 and S14, we really cannot detect consistent discrete fluorescent *QcrB* or *QcrC* dots in cells exposed to valinomycin. The membrane localization of *QcrB* and *QcrC* in untreated cells is maybe not smooth, but this is very normal for membrane proteins and is to some extent also observed for most of the other tagged membrane proteins in Fig. 6.

References

- Arce-Rodriguez, A., D. Pankratz, M. Preusse, P. I. Nikel and S. Haussler (2022). "Dual Effect: High NADH Levels Contribute to Efflux-Mediated Antibiotic Resistance but Drive Lethality Mediated by Reactive Oxygen Species." *mBio* **13**(1): e0243421.
- Azqueta, A. and A. R. Collins (2013). "The essential comet assay: a comprehensive guide to measuring DNA damage and repair." *Arch Toxicol* **87**(6): 949-968.
- Dwyer, D. J., P. A. Belenky, J. H. Yang, I. C. MacDonald, J. D. Martell, N. Takahashi, C. T. Chan, M. A. Lobritz, D. Braff, E. G. Schwarz, J. D. Ye, M. Pati, M. Vercruysse, P. S. Ralifo, K. R. Allison, A. S. Khalil, A. Y. Ting, G. C. Walker and J. J. Collins (2014). "Antibiotics induce redox-related physiological alterations as part of their lethality." *Proc Natl Acad Sci U S A* **111**(20): E2100-2109.
- Dwyer, D. J., J. J. Collins and G. C. Walker (2015). "Unraveling the physiological complexities of antibiotic lethality." *Annu Rev Pharmacol Toxicol* **55**: 313-332.
- Feld, L., G. M. Knudsen and L. Gram (2012). "Bactericidal antibiotics do not appear to cause oxidative stress in *Listeria monocytogenes*." *Appl Environ Microbiol* **78**(12): 4353-4357.
- Foti, J. J., B. Devadoss, J. A. Winkler, J. J. Collins and G. C. Walker (2012). "Oxidation of the guanine nucleotide pool underlies cell death by bactericidal antibiotics." *Science* **336**(6079): 315-319.
- Goosens, V. J., C. G. Monteferrante and J. M. van Dijl (2014). "Co-factor insertion and disulfide bond requirements for twin-arginine translocase-dependent export of the *Bacillus subtilis* Rieske protein QcrA." *J Biol Chem* **289**(19): 13124-13131.
- Guo, M., P. Tian, Q. Li, B. Meng, Y. Ding, Y. Liu, Y. Li, L. Yu and J. Li (2023). "Gallium Nitrate Enhances Antimicrobial Activity of Colistin against *Klebsiella pneumoniae* by Inducing Reactive Oxygen Species Accumulation." *Microbiol Spectr* **11**(4): e0033423.
- Hong, Y., J. Zeng, X. Wang, K. Drlica and X. Zhao (2019). "Post-stress bacterial cell death mediated by reactive oxygen species." *Proc Natl Acad Sci U S A* **116**(20): 10064-10071.
- Imlay, J. A. (2013). "The molecular mechanisms and physiological consequences of oxidative stress: lessons from a model bacterium." *Nat Rev Microbiol* **11**(7): 443-454.
- Kawai, Y., M. Kawai, E. S. Mackenzie, Y. Dashti, B. Kepplinger, K. J. Waldron and J. Errington (2023). "On the mechanisms of lysis triggered by perturbations of bacterial cell wall biosynthesis." *Nat Commun* **14**(1): 4123.
- Keren, I., Y. Wu, J. Inocencio, L. R. Mulcahy and K. Lewis (2013). "Killing by bactericidal antibiotics does not depend on reactive oxygen species." *Science* **339**(6124): 1213-1216.
- Kohanski, M. A., D. J. Dwyer, B. Hayete, C. A. Lawrence and J. J. Collins (2007). "A common mechanism of cellular death induced by bactericidal antibiotics." *Cell* **130**(5): 797-810.
- Kohanski, M. A., D. J. Dwyer, J. Wierzbowski, G. Cottarel and J. J. Collins (2008). "Mistranslation of membrane proteins and two-component system activation trigger antibiotic-mediated cell death." *Cell* **135**(4): 679-690.

Lam, P. L., R. S. Wong, K. H. Lam, L. K. Hung, M. M. Wong, L. H. Yung, Y. W. Ho, W. Y. Wong, D. K. Hau, R. Gambari and C. H. Chui (2020). "The role of reactive oxygen species in the biological activity of antimicrobial agents: An updated mini review." *Chem Biol Interact* **320**: 109023.

Liu, Y. and J. A. Imlay (2013). "Cell death from antibiotics without the involvement of reactive oxygen species." *Science* **339**(6124): 1210-1213.

Misiaszek, R., C. Crean, A. Joffe, N. E. Geacintov and V. Shafirovich (2004). "Oxidative DNA damage associated with combination of guanine and superoxide radicals and repair mechanisms via radical trapping." *J Biol Chem* **279**(31): 32106-32115.

Ohkuro, M., K. Kobayashi, K. Takahashi and S. Nagasawa (1994). "Effect of C1q on the processing of immune complexes by human neutrophils." *Immunology* **83**(3): 507-511.

Orman, M. A. and M. P. Brynildsen (2015). "Inhibition of stationary phase respiration impairs persister formation in *E. coli*." *Nat Commun* **6**: 7983.

Paulander, W., Y. Wang, A. Folkesson, G. Charbon, A. Lobner-Olesen and H. Ingmer (2014). "Bactericidal antibiotics increase hydroxyphenyl fluorescein signal by altering cell morphology." *PLoS One* **9**(3): e92231.

Ryan, T. C., G. J. Weil, P. E. Newburger, R. Haugland and E. R. Simons (1990). "Measurement of superoxide release in the phagovacuoles of immune complex-stimulated human neutrophils." *J Immunol Methods* **130**(2): 223-233.

Schafer, A. B., M. Steenhuis, K. K. Jim, J. Neef, S. O'Keefe, R. C. Whitehead, E. Swanton, B. Wang, S. Halbedel, S. High, J. M. van Dijl, J. Luirink and M. Wenzel (2023). "Dual Action of Eeyarestatin 24 on Sec-Dependent Protein Secretion and Bacterial DNA." *ACS Infect Dis* **9**(2): 253-269.

Shee, S., S. Singh, A. Tripathi, C. Thakur, T. A. Kumar, M. Das, V. Yadav, S. Kohli, R. S. Rajmani, N. Chandra, H. Chakrapani, K. Drlica and A. Singh (2022). "Moxifloxacin-Mediated Killing of *Mycobacterium tuberculosis* Involves Respiratory Downshift, Reductive Stress, and Accumulation of Reactive Oxygen Species." *Antimicrob Agents Chemother* **66**(9): e0059222.

Zhao, X. and K. Drlica (2014). "Reactive oxygen species and the bacterial response to lethal stress." *Curr Opin Microbiol* **21**: 1-6.

Zhao, X., Y. Hong and K. Drlica (2015). "Moving forward with reactive oxygen species involvement in antimicrobial lethality." *J Antimicrob Chemother* **70**(3): 639-642.

REVIEWER COMMENTS

Reviewer #1 (Remarks to the Author):

The authors have done an excellent job in addressing the comments we raised in our original review. We recommend the revised paper for publication in Nature Communications.

Reviewer #2 (Remarks to the Author):

As noted in the previous review, the claim of antibiotics killing by producing ROS is highly controversial. A source of this controversy stems from a number of factors, but all of these can be attributed to issues stemming from indirect experiments. A simple direct experiment was recommended in the previous review – examine killing under anaerobic conditions, and compare that to killing under aerobic conditions. Such an experiment was reported in two papers that the authors now cite (Keren et al., Science 2013, and Liu and Imlay, Science 2013), and the result was telling – no significant difference in killing of *E. coli* by antibiotics under aerobic vs. anaerobic conditions. The authors of this present manuscript study killing of *B. subtilis* by valinomycin, a K⁺ ionophore. This is not a great experimental system for several reasons – *B. subtilis* is not a pathogen that is treated with antibiotics, so the choice of this organism is unclear; valinomycin is not a typical target-specific compound, but an ionophore whose action has complex consequences on the cell - dissipating the membrane potential (as the authors note), but also increasing the cytoplasmic concentration of K⁺ and increasing the pH of the cytoplasm (which the authors do not note).

B. subtilis grow well under anaerobic conditions in the presence of nitrate, see for example DOI: 10.1111/j.1574-6968.1995.tb07780.x. The authors do not explain why they choose not to perform a direct, simple experiment – compare killing of *B. subtilis* by valinomycin under aerobic vs. anaerobic conditions. This is especially strange given that in both the first and second versions of the manuscript they report placing aerobically grown *B. subtilis* under anaerobic conditions (though without nitrate, so the cells die). Without a proper aerobic vs. anaerobic comparison, the many indirect experiments reported in this manuscript remain unconvincing.

Reviewer #3 (Remarks to the Author):

This is a revised manuscript that I reviewed before. The authors seem to have put a significant amount of effort into the revision, addressing many of the concerns I raised in my initial round of comments. There is no doubt that valinomycin perturbs the cell membrane

and stimulates superoxide generation, contributing to bacterial cell death. However, the authors' attempt to specifically link cell death, especially DNA damage, specifically to superoxide is problematic for the following reasons:

The authors did not specifically measure superoxide. Both H₂DCFDA and Oxyburst Green measure total ROS levels (superoxide, peroxide, hydroxyl radical), with an emphasis on hydrogen peroxide due to its much longer half-life than the other two very short-lived radicals. The oxidative fluorescent signals observed by the authors cannot distinguish superoxide from peroxide and hydroxyl radical. I suggested using MitoSox-Red, but they did not provide new data from this superoxide-specific dye, claiming that it lysed the bacteria studied. They should have attempted to titrate MitoSox-Red concentrations to find the highest dose that does not cause cell lysis to perform the experiments, as different bacteria may have varying toxicity/sensitivity to this dye.

Even if superoxide can be specifically measured and valinomycin does stimulate superoxide accumulation, the authors cannot rule out that superoxide kills, at least in part, through its dismutated products: hydrogen peroxide and subsequently hydroxyl radical. Superoxide is very unstable and can rapidly and spontaneously dismutate to hydrogen peroxide even without catalysis by SOD.

The major flaw in the current interpretation of the data is the assertion that superoxide directly damages DNA, leading to cell death. A recA mutant exhibited hypersensitivity to valinomycin, and DNA damage was detected in valinomycin-treated cells. However, there is no evidence or literature supporting the idea that superoxide can directly damage DNA. The literature cited by the authors (Misiąszek, R., JBC 2004) to support their point is flawed, as the work showed that a combination of superoxide and guanine neutral radical, not superoxide itself, can damage the guanine base. Therefore, DNA damage most likely derives from the hydroxyl radical, possibly from hydrogen peroxide (indirectly via the Fenton Reaction), as supported by previous work. The sodA mutant and the tiron data support the importance of superoxide contribution but cannot exclude its contribution indirectly through decay to hydrogen peroxide and hydroxyl radical.

The experimental conditions involving MitoSox-Red, thiourea, etc., need optimization to use compound concentrations that do not kill bacteria or inhibit bacterial growth by the compound itself. Typically, one-half or at most three-quarters of MIC concentrations should be used, as such concentrations rarely kill cells but have the best chance to display their effects. Any compound can kill bacteria or inhibit their growth when concentrations are too high, and the appropriate concentrations to be used in experiments may vary among different bacterial species. Therefore, such experiments usually require careful optimization, not simply copying published conditions used with other bacteria-drug combinations.

Without thoroughly addressing the above points in writing or even conducting new experiments, the authors' conclusions cannot be considered solid enough for publication.

Point-by-point reply to reviewers' comments

We would like to thank the reviewers for their efforts and useful comments.

To clearly indicate our replies, we have written them in italics and blue.

REVIEWER COMMENTS

Reviewer #1

The authors have done an excellent job in addressing the comments we raised in our original review. We recommend the revised paper for publication in Nature Communications.

We are happy to learn that reviewer #1 is satisfied with our additional experiments.

Reviewer #2:

As noted in the previous review, the claim of antibiotics killing by producing ROS is highly controversial. A source of this controversy stems from a number of factors, but all of these can be attributed to issues stemming from indirect experiments. A simple direct experiment was recommended in the previous review – examine killing under anaerobic conditions, and compare that to killing under aerobic conditions. Such an experiment was reported in two papers that the authors now cite (Keren et al., Science 2013, and Liu and Imlay, Science 2013), and the result was telling – no significant difference in killing of E. coli by antibiotics under aerobic vs. anaerobic conditions. The authors of this present manuscript study killing of B. subtilis by valinomycin, a K⁺ ionophore. This is not a great experimental system for several reasons – B. subtilis is not a pathogen that is treated with antibiotics, so the choice of this organism is unclear; valinomycin is not a typical target-specific compound, but an ionophore whose action has complex consequences on the cell - dissipating the membrane potential (as the authors note), but also increasing the cytoplasmic concentration of K⁺ and increasing the pH of the cytoplasm (which the authors do not note).

B. subtilis grow well under anaerobic conditions in the presence of nitrate, see for example DOI: 10.1111/j.1574-6968.1995.tb07780.x. The authors do not explain why they choose not to perform a direct, simple experiment – compare killing of B. subtilis by valinomycin under aerobic vs. anaerobic conditions. This is especially strange given that in both the first and second versions of the manuscript they report placing aerobically grown B. subtilis under anaerobic conditions (though without nitrate, so the cells die). Without a proper aerobic vs. anaerobic comparison, the many indirect experiments reported in this manuscript remain unconvincing.

We appreciate the evaluation by reviewer #2 of our manuscript, and their comments regarding the experimental design and interpretation of our results. However, as pointed out in our previous reply, anaerobic experiments do not refute our aerobic findings and conclusions and because of this, we have not further explored this route.

Reviewer #3:

This is a revised manuscript that I reviewed before. The authors seem to have put a significant amount of effort into the revision, addressing many of the concerns I raised in my initial round of comments. There is no doubt that valinomycin perturbs the cell membrane and stimulates superoxide generation, contributing to bacterial cell death. However, the authors' attempt to specifically link cell death, especially DNA damage, specifically to superoxide is problematic for the following reasons:

The authors did not specifically measure superoxide. Both H₂DCFDA and Oxyburst Green measure total ROS levels (superoxide, peroxide, hydroxyl radical), with an emphasis on hydrogen peroxide due

to its much longer half-life than the other two very short-lived radicals. The oxidative fluorescent signals observed by the authors cannot distinguish superoxide from peroxide and hydroxyl radical. I suggested using MitoSox-Red, but they did not provide new data from this superoxide-specific dye, claiming that it lysed the bacteria studied. They should have attempted to titrate MitoSox-Red concentrations to find the highest dose that does not cause cell lysis to perform the experiments, as different bacteria may have varying toxicity/sensitivity to this dye.

Our initial experiments with the MitoSox probes were indeed limited. This was due to the high price of the probe and the small quantities that we therefore acquired. We have now bought larger quantities that enabled us to test more conditions, and found, in line with the suggestion of reviewer #3, that this probe indeed can also work at lower concentrations that do not kill cells. Using 5 μ M MitoSox Red, we found the same results as with H₂DCFDA and Oxyburst, further confirming that depolarization creates superoxide radicals. We have added this new data in Fig. S6 and S12, and mention the results in the revised main text in lines 218-220 and line 320.

Even if superoxide can be specifically measured and valinomycin does stimulate superoxide accumulation, the authors cannot rule out that superoxide kills, at least in part, through its dismutated products: hydrogen peroxide and subsequently hydroxyl radical. Superoxide is very unstable and can rapidly and spontaneously dismutate to hydrogen peroxide even without catalysis by SOD.

We agree with the reviewer that we cannot rule out that hydroxyl radicals, produced from superoxide, contribute to the killing upon membrane depolarization. We have now mentioned this in lines 229-232.

The major flaw in the current interpretation of the data is the assertion that superoxide directly damages DNA, leading to cell death. A recA mutant exhibited hypersensitivity to valinomycin, and DNA damage was detected in valinomycin-treated cells. However, there is no evidence or literature supporting the idea that superoxide can directly damage DNA. The literature cited by the authors (Misiaszek, R., JBC 2004) to support their point is flawed, as the work showed that a combination of superoxide and guanine neutral radical, not superoxide itself, can damage the guanine base. Therefore, DNA damage most likely derives from the hydroxyl radical, possibly from hydrogen peroxide (indirectly via the Fenton Reaction), as supported by previous work. The sodA mutant and the tiron data support the importance of superoxide contribution but cannot exclude its contribution indirectly through decay to hydrogen peroxide and hydroxyl radical.

Guanine neutral radicals are important intermediates in the damage of DNA by ROS, including superoxide radicals, according to e.g. PMID: 34207639. However, as mentioned above, we can indeed not exclude some damage caused by hydroxyl radicals as a result of the decay of superoxide to hydrogen peroxide and hydroxyl radicals. We mention this now in lines 229-232.

The experimental conditions involving MitoSox-Red, thiourea, etc., need optimization to use compound concentrations that do not kill bacteria or inhibit bacterial growth by the compound itself. Typically, one-half or at most three-quarters of MIC concentrations should be used, as such concentrations rarely kill cells but have the best chance to display their effects. Any compound can kill bacteria or inhibit their growth when concentrations are too high, and the appropriate concentrations to be used in experiments may vary among different bacterial species. Therefore, such experiments usually require careful optimization, not simply copying published conditions used with other bacteria-drug combinations.

Following the advice of the reviewer, we have tested a range of concentrations, and we have confirmed the toxicity of thiourea in combination with valinomycin using checkerboard assays and indeed found a clear synergistic effect. We mention this now in the revision in line 249-252. In addition, we have

repeated the post-killing assay, using a range of lower thiourea and bipyridyl concentrations. The latter did not show any reduction in killing when incorporated into agar plates. A 100-fold lower thiourea concentrations (0.5 mM) showed a slight increase in CFU when incorporated in agar plates, comparable with the effect of catalase in the plates. This suggests that there is some post-valinomycin killing, but as discussed in the main text, the effect is very modest compared to what has been shown for antibiotic exposure of E. coli, and the main killing by valinomycin, therefore, does not occur after valinomycin exposure. We have added the new data in Fig. S9 and Table S5, and mention the additional controls in lines 243-244 and 248-253 of the main text.

Without thoroughly addressing the above points in writing or even conducting new experiments, the authors' conclusions cannot be considered solid enough for publication.

We hope that with the added controls and text, the reviewer will find our revised manuscript suitable for publication.

REVIEWERS' COMMENTS

Reviewer #3 (Remarks to the Author):

The authors have done a very good job in addressing all but one comments I raised in my previous round of review. The authors still overplay the direct role of superoxide to DNA damage and cell death, which has no evidence and is unnecessary. Neither the old ((Misiaszek, R., JBC 2004) nor the newly cited (PMID: 34207639) reference in their manuscript specifically demonstrates or even mentions a direct role of superoxide in damaging DNA. Indeed, to my knowledge no evidence in literature has ever demonstrated a direct damaging of DNA by superoxide. Thus, I suggest that the authors down tone this specific point. A sentence like below is more appropriate: "Since directly damaging DNA by superoxide has not been demonstrated and since superoxide is unstable and quickly dismutates to hydrogen peroxide and then hydroxyl radical by Fenton reaction, the valinomycin-mediated DNA damage is more likely caused by hydrogen peroxide or hydroxyl radical, although a direct damage by superoxide cannot be ruled out."

1 **Point-by-point reply to reviewers' comments**

2

3 **Reviewer #3 (Remarks to the Author):**

4 The authors have done a very good job in addressing all but one comments I raised in my previous
5 round of review. The authors still overplay the direct role of superoxide to DNA damage and cell
6 death, which has no evidence and is unnecessary. Neither the old ((Misiaszek, R., JBC 2004) nor
7 the newly cited (PMID: 34207639) reference in their manuscript specifically demonstrates or
8 even mentions a direct role of superoxide in damaging DNA. Indeed, to my knowledge no
9 evidence in literature has ever demonstrated a direct damaging of DNA by superoxide. Thus, I
10 suggest that the authors down tone this specific point. A sentence like below is more appropriate:
11 "Since directly damaging DNA by superoxide has not been demonstrated and since superoxide is
12 unstable and quickly dismutates to hydrogen peroxide and then hydroxyl radical by Fenton
13 reaction, the valinomycin-mediated DNA damage is more likely caused by hydrogen peroxide or
14 hydroxyl radical, although a direct damage by superoxide cannot be ruled out."

15

16 **Reply:** We thank the reviewer for the feedback and have replaced lines 229-233 by the sentence
17 suggested by the reviewer. To be precise:

18

19 "Of note, hydroxyl radicals are the most powerful oxidants among ROS and primarily responsible
20 for DNA damage. Since superoxide is unstable and can decay into hydrogen peroxide and
21 hydroxyl radical, it is likely that some of the cellular damage is caused by hydroxyl radicals."

22

23 has now been replaced by:

24

25 "Of note, since directly damaging DNA by superoxide has not been demonstrated and since
26 superoxide is unstable and quickly dismutates to hydrogen peroxide and then hydroxyl radical by
27 Fenton reaction, the valinomycin-mediated DNA damage is more likely caused by hydrogen
28 peroxide or hydroxyl radical, although a direct damage by superoxide cannot be ruled out."